# Classification of the LC4 Primarily-like Cell Line—Recapitulating a CDK4 Overexpressing Immune Evasive HIV-HCV-Induced HCC

**DOI:** 10.3390/v17050653

**Published:** 2025-04-30

**Authors:** Janine Kah, Lisa Staffeldt, Tassilo Volz, Kornelius Schulze, Asmus Heumann, Götz Rövenstrunk, Meike Goebel, Sven Peine, Maura Dandri, Stefan Lüth

**Affiliations:** 1Department of Internal Medicine, University Medical Center Hamburg-Eppendorf, 20246 Hamburg, Germany; l.staffeldt@uke.de (L.S.); t.volz@uke.de (T.V.);; 2Faculty of Health Sciences Brandenburg, Brandenburg Medical School Theodor Fontane, 16816 Neuruppin, Germany; goetz.roevenstrunk@mhb-fontane.de (G.R.);; 3Department of Gastroenterology, Center for Translational Medicine, University Hospital Brandenburg, 14770 Brandenburg, Germany; 4German Center for Infection Research, Hamburg-Lübeck-Borstel Partner Site, 38124 Braunschweig, Germany; 5Department of General, Visceral and Thoracic Surgery, University Medical Center Hamburg-Eppendorf, 20246 Hamburg, Germany; a.heumann@uke.de; 6III. Department of Medicine, University Medical Center Hamburg-Eppendorf, 20246 Hamburg, Germany; 7Institute for Transfusion Medicine, University Medical Center Hamburg-Eppendorf, 20246 Hamburg, Germany

**Keywords:** viral-induced hepatocellular carcinoma, metabolic reprogramming, drug resistance, personalized therapy

## Abstract

Background: Hepatocellular carcinoma (HCC) is one of the leading causes of cancer-related mortality. HCC is characterized by high heterogeneity and, subsequently, adaptation by developing resistance to current treatments. Applying individualized models is crucial to understanding the potential of approved therapies. Therefore, we classify a primary-like cell line derived from the core region of an HCC with underlying HIV-HCV co-infection employing deep analysis on the pathway regulation level. Methods: We employed DEG analysis, followed by pathway analysis, to characterize the preservation level of the LC4 cells and the level of adoption. Next, we classify the new model for HCC research by employing healthy donor samples, commonly used HCC cell lines, and global RNAseq datasets. Results: LC4 cells reflect the characteristics of the parental cancer region, including immunosuppression and metabolic reprogramming, characterized by the downregulation of drug-metabolizing enzymes compared to healthy individuals, indicating a transition to alternate metabolic pathways. Moreover, we identified specific biomarkers equally regulated in the parental tissue, in global datasets of the same entities as well as in LC4 cells. Conclusions: We classified LC4 cells as an individual immunosuppressive and highly progressive primary-like HCC cell line. LC4 cells are applicable as a model for preclinical drug testing, minimizing the lack of preclinical models in HCV-HIV-induced HCC research.

## 1. Introduction

Hepatocellular carcinoma (HCC) is one of the most prevalent cancers, ranking as the sixth most common malignancy and the second leading cause of cancer-related mortality [1]. Although the incidence of metabolic steatohepatitis (MASH) of alcohol- or non-alcoholic origin is increasing, chronic viral hepatitis continues to be the predominant cause of liver cancer globally [2,3]. A significant risk of developing an HCC is the infection with the hepatitis C virus (HCV), which causes chronic liver inflammation, fibrosis, and cirrhosis [4]. Of note, an additional complication in this setting represents the co-infection with the human immunodeficiency virus (HIV), whereby individuals co-infected with HIV and HCV exhibit expedited progression to liver cirrhosis and hepatocellular carcinoma facilitated by the synergistic detrimental effects of both viruses [5,6]. It has been shown that these individuals are more likely to develop resistance during therapy.

Despite the recently invented checkpoint inhibitors, the 5-year survival rate remains poor [7,8]. For late-stage HCC patients, the primary goal is prolonging survival while maintaining quality of life. In recent years, immunotherapy has emerged as a promising area, driven by advances in understanding tumor and tumor microenvironment (TME) interactions and the clinical success of immune checkpoint blockade [9]. Here, first-line therapy regimes, Atezolizumab and Bevacizumab, in combination with antibodies blocking the PD-1/PDL-1 pathway, resulted in an increased immune response [10,11,12]. However, 80–85% of the patients either do not respond or develop resistance during therapy.

The key challenges remain in elucidating mechanisms that lead to resistance and efficacy, identifying novel therapeutic targets, and developing personalized treatment regimens. To tackle these challenges, commonly used immortalized HCC cell lines, like HepG2, Huh7, and Hep3B, have significant limitations in accurately reflecting the complexity of HCC, especially in the aspects of chemoresistance and the effectiveness of targeted therapies [13]. Incorporating individual patient-derived cell culture systems has become invaluable for exploring these challenges by offering more physiologically relevant models, better reflecting tumor heterogeneity and the TME [13,14]. These models are crucial for studying the molecular pathways driving HCC progression and therapy responses. Using patient-derived cell lines from diverse etiologies, particularly those with underlying chronic viral infections and comorbidities, is necessary to mimic specific in vivo conditions, thereby enhancing the translational potential of preclinical findings to clinical applications [15]. Building on this, there is a critical need for individual models that account for variability in tumor biology and treatment responses.

In this study, we present a comprehensive analysis of the LC4 cell line derived from a patient co-infected with HIV and HCV who later developed HCC. The patient was typed as HLA-A 02:01, 03:01; HLA-B 15:01, 57:01; and HLA-C 01:02, 06:02 positive, with undetectable HIV and HCV RNA serum levels at the time of resection due to treatment with Ribavirin and Biktarvy. Our study incorporated RNA-based real-time assays, histological and cytometric protein profiling, and RNA sequencing to identify differentially expressed genes (DEGs), pathway alterations, and biomarkers to classify the LC4 cells in the HCC landscape. Our key findings show that LC4 cells exhibited distinct characteristics reflecting the parental tissue, including the highly proliferative and immunosuppressive character displayed by the upregulation of genes in the PD-L1/VEGF pathway and the downregulation of B2M, IKZF1, and CDKN2A, accompanied by CDK4 overexpression. Therefore, the established LC4 cell line reflects a clinical case of a viral HCC, showcasing immunosuppression accomplished by a poor prognosis in tumor progression [16]

## 2. Materials and Methods

Human sample collection and animal experiments. Human HCC-derived samples were collected and ethically approved as described previously [17]. Samples from male and female patients suffering from HCC between 18–99 years were collected with no demographic exclusion characteristics.

The collective of healthy primary hepatocytes was obtained after the isolation of 4 donor liver tissues which were not used for transplantation: TM14 (female, 52 years), TM47 (female, 11 months), TM46 (male, 2 months), and TM52 (female, 30 years). The studies were approved by the Ethical Review Committee of the Ärztekammer Hamburg (20359 Hamburg, Germany) (WF-021/11 and PV-3578). Handling of the human material was performed in accordance with national guidelines and the 1975 Declaration of Helsinki [18]. All animal experiments were conducted following the European Communities Council Directive (86/EEC) and were approved by the City of Hamburg, Germany (N056/2020).

HLA Typing. DNA has been extracted from patient blood cell samples for HLA Class I typing using the DNA Isolation Kit (Wizard HMW DNA Extraction Kit from Promega, GER, Walldorf, Germany). DNA was used for Luminex-based high-definition LABType rSSO typing (One Lambda, Canoga Park, CA, USA) of the HLA loci HLA-A, HLA-B, and HLA-C. LabType SSO is a reverse SSO DNA typing method (SSO) with sequence-specific oligonucleotide probes that specifically bind to homologous sequence sections of specific HLA alleles. Genomic DNA (5–60 ng/µL) was PCR-amplified using locus-specific biotinylated primers (Exons 2, 4, and 5 for HLA-A and -B; exons 2, 4, 5, 6, and 7 for HLA-C) and Amplitaq Polymerase (One Lambda, Canoga Park, CA, USA), resulting in biotinylated-amplicons. The presence of biotin in each amplicon was detectable using R-Phycoerythrin-conjugated Streptavidin (SAPE). The PCR product was denatured and hybridized to complementary DNA probes bound to fluorescently coded beads. Samples were measured by a flow analyzer (FlexMap 3D^®^, Luminex^®^, Austin, TX, USA), which identified the fluorescent intensity of phycoerythrin (PE) on each bead. The HLA FusionTM program Version 4.6.1 (One Lambda, Canoga Park, CA, USA) analyzed the data.

HCC cell culture. HCC cell lines were maintained as described previously [19,20]. For detailed information on LC4 cells, a data sheet is available in the Appendix A. For 3D conditions, different amounts of LC4 cells were seeded on BIOFLOAT™ cell culture plates in DMEM containing L-glutamine and glucose, supplemented with 1% P/S, 10% Gibco fetal bovine serum (FBS; all from ThermoFisher Scientific, Waltham, MA, USA) and were obtained for 3 weeks to follow the spheroid formation. Spheroids were used for RNA Isolation after 24 days of formation when dense and packed spheroids were obtained, as shown in Appendix A. For visualization, LC4 cells stably transduced with the vector LeGO-iG2-Puro+-Luc2 (3rd generation HIV1-derived self-inactivating vector) were used. Stable transduction was performed as described previously [17]. For 2D conditions, LC4 were seeded as described previously [17] and used for RNA isolation after forming a monolayer of over 90% density.

HCC cell culture immune cells treatment. LC4 cells, Hep3B, or HepG2.1.3 cells were seeded for two-dimensional culture into Xcelligence (Agilent) 16-well plates (10.000/well) including a negative medium control and were allowed to growth for 24 h. For T-cell experiments, LC4 cells and HepG2.1.3 cells were used in n = 5 technical replicates (3 groups; 1 = untreated; 2 = T cell treated; 3 = T cell treated + Il2 after 24 h). T-cells were reactivated with 400 IU rh-IL-2 two days before the transfer experiment. T cells were transferred in 1:1 E:T Ratio for 4 days. For the NK92 cell transfer, NK92 cells were cultured with 600 IU rh-IL-2 for several weeks. NK92 cells were transferred in different E:T Ratios (1:1, 1:5, 1:10, 1:20) and co-cultured for 3 days.

Isolation of Oligonucleotides. RNA was extracted from human liver specimens using the RNeasy Mini RNA purification kit (Qiagen, Hilden, Germany) [21]. RNA was extracted from seeded cells with or without treatment after indicated time points using the RNeasy RNA Micro purification kit (Qiagen, Hilden, Germany).

Measurement of gene expression level using TaqMan-based PCR. For measurement of gene- expression, two-step PCR was performed. Therefore, complementary DNA (cDNA) synthesis was conducted using MMLV Reverse Transcriptase 1st-Strand cDNA Synthesis Kit (Lucigen, Middleton, WI, USA) to synthesize RNA complementary DNA, according to the manufacturer’s instructions. Human-specific primers from the TaqMan Gene Expression Assay System, listed in Appendix A, were used to determine gene expression levels (Life Technologies, Carlsbad, CA, USA). Samples were analyzed using the Quant Studio 7 Real-Time PCR System (Life Technologies, Carlsbad, 92008 CA, USA). The human housekeeping genes, glyceraldehyde-3-phosphate dehydrogenase (GAPDH) and ribosomal protein L0 (RPL0), were used to normalize human gene expression levels.

Flow cytometry. LC4 cells line LC4 were characterized using the monoclonal antibodies anti-CD68-APC700 and anti-CD45-BV510. Cells were stained as described previously, and measurement was carried out on the BD FACSymphony™ A3 Cell Analyzer from BD Biosciences (Heidelberg, Germany).

Protein analysis by immunofluorescence. For histological characterization, cultured cells and tissue slides were processed as described previously [17,22]. Antibodies used in the study are listed in Appendix A. Stained cells and tissue slides were analyzed by fluorescence microscopy (BZ-9000 and BX-780, Keyence, Osaka, Japan). Captures were automatically generated with the indicated magnification and consistent exposure for the same staining.

RNA Isolation and Sequencing. Total RNA was extracted from primary hepatocyte samples, the primary HCC cell line LC4, and immortalized HCC cell lines (HepG2, Huh7, and Hep3B) using the RNeasy Mini Kit (Qiagen, 40724 Hilden, Germany), following the manufacturer’s protocol. The quality and quantity of RNA were assessed using a NanoDrop spectrophotometer (ThermoFisher Scientific, Karlsruhe, Germany) and an Agilent Bioanalyzer 2100 (Agilent Technologies, Waldbronn, Germany), using the High Sensitivity DNA Chips (Cat: 5067-4626). High-quality RNA samples with an RNA Integrity Number (RIN) greater than 7.0 were selected for sequencing.

Library Preparation and RNA Sequencing. RNA sequencing libraries were prepared using the TruSeq Stranded mRNA Library Prep Kit and the index adapter kit IDT-Limn RNA UD Indexes set a Ligation according to the manufacturer’s instructions (Illumina, 81669 Munich, Germany). MRNA was purified from total RNA using poly-T oligo-attached magnetic beads and fragmented into small pieces. First-strand cDNA was synthesized using random hexamer primers and reverse transcriptase. This was followed by second-strand cDNA synthesis, end repair, A-tailing, adapter ligation, and PCR amplification to enrich the cDNA fragments. The libraries were quantified using a Qubit fluorometer (ThermoFisher Scientific, Karlsruhe, Germany), using the Qubit 1X dsDNA HS Assay Kit (Q33230) and assessed for size distribution using the Agilent Bioanalyzer 2100 (Agilent Technologies, Waldbronn, Germany). The libraries were then sequenced on the Illumina NextSeq 1000/2000 platform (Illumina, Munich, Germany, generating 200 bp paired end reads. Technical replicates were used for RNAseq for each sample (n = 2).

Data Processing and Differential Expression Analysis. Raw sequencing reads (n = 2 per sample) were prepared as R input datasets and further processed with R [23]. Input was trimmed using Trimmomatic [24]. Clean reads were aligned to the human reference genome (GRCh38) using a STAR aligner [25]. The resulting BAM files were processed with featureCounts to generate read count matrices [26]. Differential expression analysis was performed using the DESeq2 package in R [27]. Genes with an adjusted *p*-value (Benjamini–Hochberg correction) of less than 0.05 and a log2 fold change (log2FC) greater than 2 or less than −2 were considered significantly differentially expressed. Detailed information is described in the Appendix A.

Data Processing and Visualization. DEGs were used for volcano plot generation, processing comparison analysis in IPA for pathway comparison, and in silico analysis of molecules as biomarkers or drug targets in IPA. Results from IPA comparisons were used for bubble plots, z-score-based heatmaps, and correlation heatmaps. The data processing, comparison, and illustration of correlation and pathway comparisons are described in detail in the Appendix A [28,29,30,31].

## 3. Results

### 3.1. The Parental Tumor Regions Are Classified as Immunogenic Margins and Immunosuppressive Core 

First, we conducted a detailed analysis of spatial heterogeneity of the parental HCC tissue, comparing the core and margin regions using expression profiling and immunofluorescence staining. As shown in Figure 1A, the hepatocyte-specific genes like CLRN3, AADAC, and ALB decreased in the core region, indicating the transformation from healthy hepatocytes to tumorigenic cells. 

GLUT1 and HIF1a were elevated in the margin, suggesting stemness, metabolic adaptation, and hypoxic conditions linked to aggressive tumor biology and potential invasiveness. As shown in Figure 1A,B, the proliferation marker Ki67 was upregulated in the core region compared to the margin, which shows signals of hypoxic adaptation and invasion. The protein levels shown in Figure 1B further illustrate these spatial differences, with CD44 and Vimentin (VIM) being more prominent in the margin, highlighting the epithelial–mesenchymal transition (EMT) at the invasive front. CD68, a macrophage marker, is present in the margin and core, indicating tumor-associated macrophage (TAM) infiltration throughout the tumor. However, the roles of these macrophages may vary in the margin, as they promote inflammation and tissue remodeling, while in the core, they contribute to immune evasion through elevated PD-L1 expression. The presented CD3-positive T-cell infiltration in the margin indicates a more active immune response, whereas higher PD-L1 expression in the core refers to a consistent immune evasion mechanism.

The higher occurrence of CD3 positive T cells In the margin, detected by gene expression levels shown in Figure 1C and supported by the protein level shown in Figure 1B, was in line with the higher expression of the immunosuppressive cytokine IL10 in the core region. The detection of elevated TNFα in the margin revers to ongoing inflammation at the tumor border. Figure 1D shows a higher CD4/CD8 ratio in the margin, indicating a stronger T-cell response.

Figure 2A examines the spatial distribution of the HCV core protein (Genotype 1b) and the co-localization of CD3-positive T-cells. As shown in the upper panel, HCV core protein co-localizes with Calnexin, a hepatocyte marker, exclusively in the core of the parental tissue. Even though CD3-positive T-cells were most prominent in the margin, confirming once again the immunosuppressive environment for the core, we found a clear interaction of CD3-positive T-cells with HCV core GT1b positive cells, as shown in the higher magnification lower panel of the Figure 2A. The distinct distribution of CD3-positive T-cells between the margin and core draws the spatial heterogeneity of the tumor, where the core, harboring viral proteins, is more adept at suppressing immune infiltration, while the margin maintains stronger immune activity.

In Figure 2B, we provided a comprehensive view of the differential gene expression (DEG) between the core and margin. In line with our previous findings, genes involved in antigen presentation, such as HLA-B, HLA-C, and HLA-DRB5, are significantly higher expressed in the margin, reflecting the enhanced immune recognition at the periphery compared to the center. The downregulation of metabolic genes and the upregulation of IGFBP3 in the core shows a shift toward proliferation and immune evasion, as shown before. The lower infiltration of CD3-positive T-cells in the core aligns with these expression patterns, further supporting that the tumor core adopts immune-suppressive strategies. Taken together, the tumor tissue from the patient reflects the common picture of most HCCs, whereby the margin is more inflammatory and invasive, while the core exhibits higher proliferation and immune evasion.

For the generation of the LC4 cell line, we employed isolated cells from the core region. Based on this, we aimed to classify the preservation of core region specific characteristics in the LC4 cell line, as environmental changes occur after cultivation. The DEG comparison reported in Figure 2C shows a clear shift in the genetic composition due to isolation procedure. In the tumor core, cells face hypoxic conditions, limited nutrient supply, and immune suppression, which keep certain genes downregulated, as shown in Figure 2B. After the isolation procedure, the core derived cells have been exposed to a more favorable environment with improved oxygen and nutrients, whereby stress responses, metabolic reprogramming, and activation of pathways associated with survival, proliferation, and invasion were triggered. Additionally, isolation simulates wound-healing or tissue-regeneration responses and activating genes involved in extracellular matrix remodeling and immune modulation as mentioned in Figure 2C. As in Figure 2D,E, we summarized the transcriptomic status of the core (D) and the margin (E) in the context of healthy hepatocytes; we found the margin with a close-to-healthy region, with only some downregulated genes involved in inflammation and metabolic pathways. On the other hand, the core region represents the same profile as described before with highly significant upregulation of genes involved in cell migration, cytoskeletal restructuring, and immune modulation. In the core, we found GPC3 downregulated, while CD24 was upregulated. Genes like TPM4, TGM2, and S100A11 were significantly upregulated in the core, reflecting enhanced cell migration and invasiveness, consistent with the findings in Figure 1. The downregulation of lipid metabolism-related genes like CYP17A1 and SQLE suggests the core reconfigures its metabolic pathways to support aggressive tumor behavior, focusing more on structural adaptation and motility. While showing signs of tumor adaptation, the margin retained normal tissue characteristics, with downregulation of genes involved in iron accumulation, detoxification, and immune response (FTL, SERPINA3, IGHA1). This aligns with earlier observations that the margin shows a less aggressive, more regulated state due to the presence of an active immune response.

### 3.2. The Generated LC4 Cell Line Positively Correlates with the Parental HCC Core Region in Pathway Analysis and Shows High Similarity to Liver-Cancer-Related Disease

To investigate the preservation of the parental characteristics and to classify patient-derived LC4 cells, we performed DEG analysis from LC4 cells in 2D and 3D culture condition to healthy hepatocytes, to the parental tissues (core and margin) and to the cell isolate derived from the HCC core region, which was used for the generation of the LC4 cell line. Therefore, we performed DEG combinations of all specimens and used the results to acquire the differential affected canonical and toxicity pathways. We next extracted the associated z-scores from the comparison from the Pathway finder software (IPA) and calculated the pairwise Pearson correlation, plotted in Figure 3A, whereby the right site represents the canonical and the left side the toxicity pathways. The correlation heatmap matrix visualizes the strength and direction of correlations between the different specimen comparisons, as mentioned in the legend. When correlating the HCC core tissue compared against the margin (X3) or the healthy hepatocytes (X27), we found a positive correlation in both canonical and toxicity pathways. The matrix also clearly shows a positive correlation between both pathway regulations detected in in the LC4 cells when compared to healthy hepatocytes (X31, X32) or the tumor margin tissue (X53, X54) with the HCC core tissue when compared to the tumor margin tissue (X3). When comparing the HCC core tissue against the healthy hepatocytes (X27), a positive correlation was detected to LC4 cells (2D and 3D) compared against the HCC core (X48, X49) and margin (X53, X54) to the same intent. As expected, the isolated HCC core cells when compared to healthy hepatocytes (X30) showed a negative correlation to both LC4 cells (X31, X32) and the parental tissue (X27). This negative correlation was even stronger in the toxic pathways. Interestingly, the correlation of the canonical pathways differs from the toxicity pathways to some intent, especially when correlating the parental margin and core both compared to healthy hepatocytes (X26, X27). Here, both regions showed a positive correlation in the canonical pathways, but a negative correlation in the toxicity. As expected, LC4 cells showed a positive correlation between the comparisons against healthy hepatocytes, core tissue, and the margin and a weak or negative correlation when compared with the isolated cell fraction (X50, X51). Taken together, the LC4 cell line in 2D and 3D conditions were able to reflect the canonical and toxicity pathways regulation as detected in the parental core tissue. To underline these findings, we depicted the underlying volcano plots of the DEGs for X53 (B) and X54 (C) in Figure 3. Here, the differential gene expression shown in Figure 3B,C reflect the immunosuppressive character of the parental tumor core region, by downregulation of genes involved in immune response, inflammation, and extracellular matrix remodeling. To classify the generated cell lines in a global setting, we employed the IPA analysis match function. Here, we filtered the four datasets and analysis from human liver and set the overall z-score to 10. The highest similarities have been plotted in Figure 3D,E, whereby the highest number of studies which show similarity to our analysis was in the HCC in both conditions. These results underline the previously shown results and help to classify the LC4 cell line as HCC representative cell line.

### 3.3. LC4 Cell Cines Retain Core Tumor Traits but Exhibit Enhanced Proliferation and Reduced Stress-Related Pathways Compared to Parental HCC Core

Based on the correlation matrix, shown in Figure 3A, we focused on a deep pathway activity analysis of parental HCC core tissue (FC3) with LC4 cell lines derived from it, grown in both 2D (FC53) and 3D (FC54) cultures. As shown in Figure 4A, LC4 cells in both 2D and 3D culture display similarities in metabolic pathways to the parental HCC core tissue. However, LC4 cells show enhanced activity in pathways related to transcription, protein processing, and translation, as shown in Figure 4B. This shows that LC4 cells, particularly in a 3D culture, exhibit a partially higher biosynthetic demand, reflecting the adaptation to in vitro conditions that differs from the in vivo environment. As summarized in Figure 4B, higher activation of metabolic pathways was detected in LC4 cells under different conditions, indicating that LC4 cells shifting their metabolic activities from the stress-adapted metabolic profile in the core. As shown in Figure 4C, LC4 cells displayed elevated activity in pathways associated with viral infections such as viral hepatitis (8), inflammation of the liver (7), and hepatitis B infection (1). This shows that LC4 cells recapitulate aspects of immune and viral stress pathways, possibly due to changes in the microenvironment under in vitro conditions. As expected, the liver tumor pathway (10) is more activated in the parental HCC core than in the LC4 cells.

LC4 cells, particularly in 3D culture, demonstrate higher predicted activity in proliferation of hepatic stellate cells (4) and liver cells (3) compared to the HCC core, due to the absence of growth-inhibitory signals or stress factors present in the tumor microenvironment. Additionally, in Figure 4C, the comparison displayed higher z-scores for pathways related to viral infection, cell proliferation, survival, and cell movement, highlighting their more active state in processes that are critical for tumor growth and metastasis. This finding points out that the 3D condition of LC4 culture reflects a better model for studying aggressive cancer behavior compared to the 2D culture. Conversely, the HCC core tissue displays higher activation of pathways associated with necrosis (9), sensitivity (2, 3), and apoptosis (10), showing that under in vivo conditions, an increased stress level leads to increased cell death and tissue damage. These pathways are significantly downregulated in LC4 cells, implying that the cultured cells are more resistant to stress and apoptotic signals, likely due to the controlled environment of in vitro culture.

Taken together, the LC4 cell line, and particularly under 3D culture conditions, aligns with the parental HCC core in terms of overall pathway activity. While we could show that LC4 cells exhibit enhanced transcriptional, protein processing, and proliferative activities, reflecting their adaptation to in vitro conditions, the parental HCC core tissue shows higher activation of tumor-specific and stress-related pathways, such as necrosis and apoptosis.

### 3.4. LC4 Patient-Derived Cells Retain Immune Evasion and Proliferative Traits but Show Reduced Immune Marker Expression Compared to Parental HCC Core

To explore the adoption and preserved characteristics of the patient-derived LC4 cells in terms of liver specific and immune specific properties, we perform a detailed comparison using the parental tissue, the isolated cells, and the generated cell lines. We aimed to determine whether the LC4 line maintains the diverse cellular components of the parental center region and if specific populations, particularly tumor associated immune cells, as TAMs are diminished during culture establishment. As shown in Figure 5A, liver- and tumor-associated markers, such as AADAC, CLRN3, GPC3, CD40, and ALB, were downregulated in LC4 cells compared to freshly isolated tumor core cells and parental tissues. In contrast, the marker CD47 was consistently expressed, and CD44 appeared to be upregulated in comparison to the parental core region. The persistent expression of CD44 and CD47, linked to stemness and immune evasion, suggests these properties are retained in LC4 cells. In Figure 5B, a clear reduction in gene expression of immune cell markers for identifying TAMs and MDSCs (e.g., CD68, ARG1, CD163, and S100A8/A9) was detectable in LC4 cell conditions, whereby the markers CD163 and ITGAM were upregulated under 3D culture conditions. The cytometry analysis shown in Figure 5C points out that a minor subset of the LC4 cells (11%) presents CD68 at very low levels. As shown by immunofluorescence data, represented in Figure 5D, LC4 cells retained markers which further support the result that LC4 cells preserve the proliferative and immune-evasive properties (e.g., Ki67, PDL-1, and CD44) of the parental tumor core. Next, as shown in Figure 5E,F, we analyzed MHC class I, class II molecules and Hif1a in the parental samples compared to their cultured counterparts. We found that HIF1a and HLA-G were upregulated in expression, while HLA-F was expressed to the same intent as in the core region. MHC class I molecules HLA-A, B and C were expressed to lower amounts, but detectable, while the expression pattern of MHC-class II molecules appeared steadily reduced in the LC4 cells.

### 3.5. LC4 Cells in 3D Culture Exhibit Enhanced Proliferation and Immune Evasion

To better characterize the newly generated, patient-derived LC4 cell line, we conducted a comprehensive differential gene expression (DEG)-based pathway analysis, comparing LC4 cells to healthy hepatocytes. As shown in Figure 6A,B, volcano plot visualizations of DEG results laid the foundation for our pathway analysis. For LC4 cells, Figure 6A confirmed findings from Figure 4, showing upregulation of genes related to protein synthesis and signal transduction (e.g., RPL30, SPTAN1), reflecting their adaptation to artificial conditions that promote rapid growth. However, the downregulation of genes involved in the extracellular matrix, metabolism, and immune response (e.g., COL3A1, NNMT, C4A, HLA-A) was observed. Figure 6B revealed that 3D-cultured LC4 cells more closely retained key characteristics of the parental tumor core when compared to healthy hepatocytes. In 3D culture, LC4 cells upregulated genes involved in cytoskeletal dynamics (e.g., CORO1B, CCT7) and immune evasion (e.g., PDL1, JAK2), suggesting that 3D conditions better replicate the tumor core’s cellular environment. Interestingly, the downregulation of genes related to immune response and lipid metabolism (e.g., CFH, SCARB1) in 3D culture suggests that some aspects of the tumor microenvironment, including immune modulation, are maintained. In the in-silico z-score-based pathway analysis, presented in Figure 6C,D, we compared LC4 cells cultured in 2D and 3D conditions with healthy hepatocytes to further explore pathway activations and inhibitions. As shown in Figure 6C, in LC4_2D culture, pathways such as neutrophil degranulation, EIF2 signaling, and macrophage production of nitric oxide and reactive oxygen species were upregulated, indicating an inflammatory milieu. In line with findings from Figure 5, MHC class II proteins were downregulated, reflecting immune modulation and stress adaptation, even compared to healthy hepatocytes. Additionally, EIF2 signaling, a key pathway responding to cellular stress, showed high activation. However, pathways related to eukaryotic translation elongation and protein ubiquitination were downregulated, suggesting reduced protein synthesis and post-translational modification in 2D culture, diverging from the patterns observed in the parental tumor tissue. In Figure 6D, LC4_3D cultures showed increased activity in pathways related to neutrophil degranulation, EIF2 signaling, and mitotic metaphase and anaphase, indicating an inflammatory environment, enhanced protein synthesis, and increased cell division. The upregulation of mitotic pathways highlights the highly proliferative nature of tumor cells in 3D culture, which better mimics the spatial structure of the parental tumor. However, pathways such as protein ubiquitination, class I MHC antigen processing and presentation, and the RHO GTPase cycle were reduced, suggesting impaired immune recognition and altered cytoskeletal regulation.

### 3.6. LC4 Cells Exhibit Stronger Transcriptomic Changes Compared to Common Liver Cancer Cell Lines, with 3D Culture Retaining More Tumor-like Characteristics

In the analysis of gene expression related to the viral hepatitis transcriptome, shown in Figure 6D, and the tumor environment transcriptome, shown in Figure 6E, LC4 cells (in both 2D and 3D culture) demonstrate distinct patterns compared to commonly used liver cancer cell lines (HUH7, HepG2, Hep3B). HUH7, HepG2, and Hep3B show relatively stable and minimal changes for the viral hepatitis-related transcriptome, with expression levels close to those of healthy samples (HD). In contrast, LC4 cells, particularly in 2D culture, display broader expression changes and more substantial downregulation of viral hepatitis-related transcripts than the commonly used cell lines. This suggests that LC4 cells, especially in 2D, undergo more significant suppression of viral stress or inflammation-related pathways. LC4_3D exhibits a similar trend but with less pronounced downregulation, suggesting that 3D culture conditions help preserve some viral hepatitis-related characteristics. For the tumor-environment-related transcriptome, the commonly used cell lines (HUH7, HepG2, Hep3B) show minor changes, with expression levels remaining close to those of healthy donors, suggesting a limited alteration in tumor-environment-related pathways. In contrast, LC4_2D and LC4_3D show broader and more significant downregulation of tumor-environment-related transcripts, with LC4_2D exhibiting the most potent reduction. This indicates that LC4 cells, particularly in 2D culture, lose some environment-specific tumor features in vivo. LC4_3D, on the other hand, retains more environment-related tumor characteristics compared to 2D culture, as indicated by less severe downregulation, making 3D culture a better model for replicating the tumor’s original microenvironment. In direct comparison, LC4 cells demonstrate more significant transcriptomic changes, particularly in 2D culture, than the commonly used liver cancer cell lines.

### 3.7. LC4 Cells Present a Lower Susceptibility Towards Active Allogenic T-Cells and Cytolytic NK Cells In Vitro

To better understand and confirm the findings acquired by next generation sequencing followed by in-depth pathway analysis, we performed a proof-of-concept by treating the LC 4 cells on the one hand with activated T cells from an allogenic donor with HLA-A02:01 and on the other hand with immortalized activated NK92 cells. For a direct comparison and as positive control, we employed HepG2.1.3 cells which display the same haplotype as LC4 cells for the T cell transfer and Hep3B cells as the MICA/B-expressing cell line for the NK92 cell transfer. Both experiments were performed using the same settings, cell numbers, activation times, and treatment time points, as displayed in Figure 7A,B. In Figure 7C,E, the transfer of active allogenic T cells was observed over a period of 4 days. The highly significant reduction of the cell index was only measurable when transferring T cells to the susceptible control cell line HepG2.1.3. The cytolytic effect was even stronger after additional application of IL-2 after 24 h, as shown in Figure 7C. For LC4 cells, we observed a much lower cytolytic effect, even when application of exogenous IL-2 was performed, as shown in Figure 7E. Another immune cell transfer experiment using LC4 cells is displayed in Figure 7B,D,F. NK92 cells were used in different E:T ratios for the transfer to the susceptible HCC cell line Hep3B and to LC4 cells. The cytolytic effect was highly induced when applying 1:1 E:T NK92 cells on Hep3B cells, while this was not the case when treating the LC4 cells. Here, NK92 cells exhibit a strong cytolytic effect at higher E:T Ratios, like 1:10 and 1:20 E:T ratios but not a 1:1 E:T ratio, as shown in Figure 7F.

### 3.8. CYP Enzyme Expression in LC4 Reflects Metabolic Reprogramming in HCC

The CYP enzyme expression, shown in Figure 8, provides key insights into drug metabolism, chemotherapy resistance, and hormonal imbalances in the LC4 model. Enzymes, essential for metabolizing drugs, steroids, and hormones, show significant differences in the expression level of LC4 cells compared to hepatoma cell lines HUH7, HepG2, and Hep3B. Figure 8A shows the downregulation of CYP3A4, CYP3A5, and CYP2C9, crucial for metabolizing chemotherapeutics like paclitaxel and docetaxel. This suggests higher drug concentrations, increased toxicity, and diminished activation of prodrugs, contributing to chemotherapy resistance. Figure 8B reduced CYP19A1 and CYP17A1 expression profiles, impacting estrogen and androgen pathways, were detected in LC4 cells. This imbalance can promote tumor growth and reduce the effectiveness of hormone-based therapies, such as aromatase inhibitors. The disruption in steroid hormone biosynthesis likely affects tumor progression and response to hormone therapies. LC4 cells also uniquely express CYP51A1 and CYP4B1 (Figure 8C,D), enzymes linked to cholesterol biosynthesis and xenobiotic metabolism. CYP51A1 suggests alternative metabolic pathways, while CYP4B1 enhances detoxification, potentially increasing chemotherapy resistance by inactivating drugs more effectively. In Figure 8E, CYP downregulation in LC4 cells (CYP2C9, CYP2D6, CYP1A1) mirrors patterns seen in HCC patient samples, especially in patients co-infected with HIV and HCV. This similarity reinforces LC4’s relevance as a preclinical model, accurately reflecting metabolic deficiencies seen in patients. The unique expression of CYP51A1 and CYP4B1 in LC4 cells suggests the model may capture additional metabolic reprogramming, offering insights into drug resistance mechanisms.

### 3.9. Biomarker Identification in LC4 Cells Reveals That CDKN2A, IKFZ1, and B2M Are Consistently Regulated

Figure 9 compares biomarker profiles across HCC core, LC4_2D, and LC4_3D cultures, revealing both conserved molecular characteristics and condition-specific adaptations. Figure 9A illustrates the overlapping of investigated biomarkers. In Figure 9B, the biomarker filter process is described. Here, we reduced the markers to a relevant group for diagnosis and efficacy observations. In this process, we identified 52 key biomarkers that were consistently expressed, emphasizing those that could impact therapeutic efficacy and serve as drug targets. Figure 9C categorizes these biomarkers into 14 consistently downregulated and five consistently upregulated markers. The heatmap in Appendix A further gives a more detailed picture of the regulated biomarkers. To understand the broader impact of the filtered biomarkers, a global dataset from the IPA platform was employed to compare them with HCCs from various entities. As a result, we identified biomarkers—IKZF1, B2M, and CDKN2A—which showed equal expression profiles along all analyzed specimens.

As shown in Appendix A, IKZF1 is crucial to pathways involved in immune regulation, such as Th2 signaling, B cell receptor signaling, and JAK/STAT pathways. Its downregulation in LC4 cells reflects an immune suppressive milieu, impairing lymphocyte development and contributing to tumor immune evasion. These findings align with our previous observations, where LC4 cells displayed reduced immune function and enhanced immune evasion. Appendix A focuses on B2M, a key component of the MHC Class I complex crucial for antigen presentation. The downregulation of B2M in LC4 aligns with the results presented in Figure 5 and Figure 7. The reduced antigen presentation impairs cytotoxic T-cell responses. This downregulation is consistent with the immune-suppressive environment of the parental tissue, particularly in the context of PD-1/PD-L1 signaling.

In Appendix A, CDKN2A, a tumor suppressor involved in cell cycle regulation, is downregulated in LC4 cells. The loss of CDKN2A function allows activation of the ERK/MAPK and HIF1α signaling pathways, which promotes cell proliferation and survival under hypoxic conditions. The downregulation of CDKN2A mirrors the behavior observed in HCCs, where the loss of cell cycle control facilitates unchecked tumor growth.

### 3.10. LC4 Cells Offer a Model for CDK6-Independent Regulation of CDKN2A Expression

In Figure 9E, CDKN2A, CDK4, and CDK6 expression levels were compared across different patient groups, including patients with cirrhosis, HCV-related HCC, non-HCV-related HCC, and parental tissue. LC4 cells display a notable downregulation of CDKN2A and upregulation of CDK4, like the molecular profile of HCV-related HCC patients. This shows that LC4 cells mimic the CDKN2A-CDK4/6 axis of viral HCCs. Interestingly, this expression pattern differs from that observed in immortalized cell lines, where CDK4 is upregulated, but CDKN2A and CDK6 exhibit varying expression levels. The consistency of LC4 cells with patient-derived HCC samples highlights their relevance as a model for studying therapeutic responses. The ailment with patient data points to the value of LC4 cells in investigating not only CDK4/6 inhibition but also their interaction with immunotherapies or anti-fibrotic treatments, given the downregulation of B2M and IKZF1.

## 4. Discussion

Our study underscores the complexity of HCCs by examining the spatial heterogeneity within a single tumor’s margin and the core region. The classification of the patient-derived LC4 cells exhibits a high similarity to the parental region. Our findings show that LC4 cells can mirror the clinical aggressive and immunosuppressive status of HIV-HCV-induced HCCs in vitro. Moreover, in a global match analysis against LC4 cells, the highest similarity in the bio function and canonical pathway changes compared to healthy donors were detected in HCC and HCC-related diseases.

We offered a complex comparison of different conditions and tissue regions to extract a DEG-based classification for a newly generated primary-like cell line in a viral HCC context. Comparing the parental tumor core and margin, distinct molecular profiles were revealed, with the tumor margin showing elevated expression of markers such as CD44 and HIF1A, consistency with stemness, hypoxic adaptation, and metabolic reprogramming at the invasive front. In contrast, the core region exhibited a high proliferation rate, as indicated by Ki67 upregulation. These features align with the LC4 cells in vitro, which showed elevated expression of CD44 and Ki67, along with significant cell cycle dysregulation, recapitulating the dynamics observed in viral-induced HCCs [32]. The upregulation of CDK4 and downregulation of CDKN2A in LC4 cells further reflect the aggressive phenotype commonly found in patients suffering from viral-related HCCs [33], making LC4 cells a suitable model for investigating the efficacy of CDK4/6 inhibitors in combination therapies [34]. A critical feature of HCC, particularly viral infections, is metabolic reprogramming. We observed significant downregulation of CYP enzymes (e.g., CYP3A4, CYP2C8) in LC4 cells, a hallmark of chemotherapy resistance [35]. Reduced expression of these enzymes impairs drug metabolism, leading to higher drug toxicity and diminished activation of prodrugs [36,37]. These findings align with previous results, showing that HepG2 cells, overexpressing CYPs, exert lower resistance to chemotherapy [17]. Interestingly, the upregulation of CYP51A1 in LC4 cells points to an alternative metabolic pathway involving cholesterol biosynthesis. Cholesterol metabolism has been increasingly implicated in tumor survival and proliferation [38]. The fact that LC4 cells clearly adopt this pathway, and the parental HCC cells were generally characterized as LXR/RXR activating, suggests that inhibitors of cholesterol biosynthesis, such as statins, could be explored as an adjunct therapy to reduce tumor growth, in this cell line. This emphasizes the importance of LC4 cells for investigating the link between metabolic reprogramming and drug resistance.

One of the most significant observations in LC4 cells is their ability to model the immune-suppressive environment found within the parental tumor, mainly through the downregulation of B2M, a critical component of the MHC Class I antigen presentation pathway, and the inhibition of IKZF1, a transcription factor involved in lymphocyte recruitment, highlighting key mechanisms of immune evasion. The reduction in B2M expression reflects a well-known mechanism of immune evasion, where tumor cells reduce their visibility to cytotoxic T cells, helping them evade immune detection [39]. This is particularly relevant in virus-induced HCC, where chronic immune stimulation from infections drives immune exhaustion and suppression [40]. The downregulation of B2M in LC4 cells mirrors what has been observed in clinical HCC cases, where loss of antigen presentation is associated with resistance to immune checkpoint inhibitors, such as PD-1/PD-L1 therapies [41,42]. The potential for LC4 cells to model immune evasion strategies makes them a valuable preclinical tool for investigating combination therapies that could overcome resistance. For instance, treatments aimed at restoring MHC Class I expression could be combined with checkpoint inhibitors to boost the immune system’s ability to recognize and destroy tumor cells. This strategy has shown potential in preliminary studies. Additionally, the observed dysregulation of IKZF1 further supports the utility of LC4 cells for studying the effectiveness and resistance of immune-modulatory therapies [43]. Given the interplay between chronic viral infection and immune suppression, LC4 cells offer a promising platform to explore novel immunotherapeutic strategies that combine a checkpoint blockade with agents to restore T-cell function.

While LC4 cells are a valuable tool for studying HCC biology, they also present limitations, particularly when comparing 2D and 3D culture models. LC4_3D cultures, which better mimic the in vivo tumor microenvironment, offer advantages in studying tumor–stroma interactions, which are essential for understanding tumor progression and therapeutic responses [44,45]. The spatial organization captured in 3D cultures makes them more physiologically relevant, especially regarding migratory and persistence-related research. Although they do not fully replicate tumor–stroma dynamics, 2D cultures remain useful for investigating tumor-intrinsic properties, such as proliferation and cell cycle dysregulation. For instance, the dysregulation of the CDKN2A-CDK4/6 axis in LC4 cells makes the 2D model particularly suited for preclinical testing of CDK4/6 inhibitors, allowing focused analysis of tumor cell behavior in a controlled environment free from stromal influences [10].

Despite the classification as an HCC-derived cell line by comparing the global analysis, the consistent regulation of key biomarkers such as CDKN2A, IKZF1, and B2M in LC4 cells and across various specimens underscores their ability to reflect specific entities of HCC. Furthermore, the insights into chemotherapy resistance and tumor–stroma interactions, particularly in 3D culture systems, offer valuable perspectives for future research and treatment development.

In conclusion, our study provides a comprehensive analysis of the spatial heterogeneity within a viral-induced HCC, highlighting the value of LC4 patient-derived cells as a model for investigating tumor progression, immune evasion, and therapeutic resistance. The low CYP expression and the immunosuppressive characteristic offers the possibility to perform drug screening assays in a clinically relevant setting for a better translation of preclinical results.

## Figures and Tables

**Figure 1 viruses-17-00653-f001:**
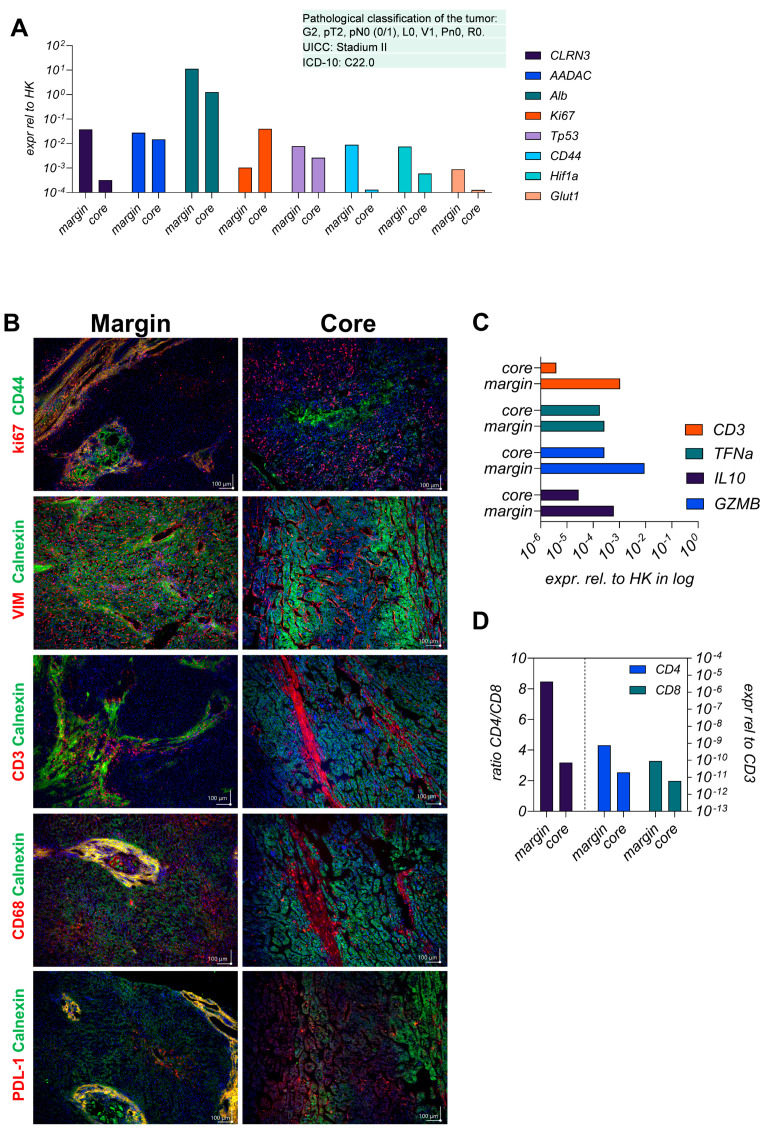
Analysis of gene and protein expression levels in patient-derived tumor tissue regions. (**A**) displays mRNA expression levels of indicated genes resented in the margin and core region of the tumor. Gene expression was normalized relative to HK gene expression (GAPDH and RPL30) and represented on a log10 scale. The analyzed genes include CLRN3, AADAC, Alb, Ki67, Tp53, CD44, Hif1a, and Glut1. The tumor is classified according to pathological criteria: G2, pT2, pN0 (0/1), L0, V1, Pn0, R0. UICC: Stage II, ICD-10: C22.0. (**B**) represents results in 20-fold magnification from immunofluorescence staining of the resected tumor tissue comparing the margin and core region. The top row displays the localization of Ki67 (red) and CD44 (green) in both the margin and core, revealing differences in proliferative activity and stemness between these areas. Subsequent rows depict staining of VIM (red) and Calnexin (green), CD3 (red) and Calnexin (green), CD68 (red) and Calnexin (green), and PDL-1 (red) and Calnexin (green). Scale bars represent 100 µm distance. (**C**) presents mRNA expression levels of immune-related genes, including CD3, TNFα, IL10, and GZMB, in the HCC4 tumor core and margin. Expression levels are normalized relative to housekeeper RPL0 gene expression and shown on a logarithmic scale. This comparison illustrates variations in immune response gene expression between the core and margin of the tumor, suggesting differential immune activity within the tumor microenvironment. (**D**) shows the ratio of CD4/CD8 T cells and the expression levels of CD4 and CD8 mRNA in the HCC4 tumor margin and core relative to the T cell lineage marker CD3. The data are presented on a logarithmic scale, highlighting the differences in T cell populations and their gene expression between the margin and core regions of the tumor.

**Figure 2 viruses-17-00653-f002:**
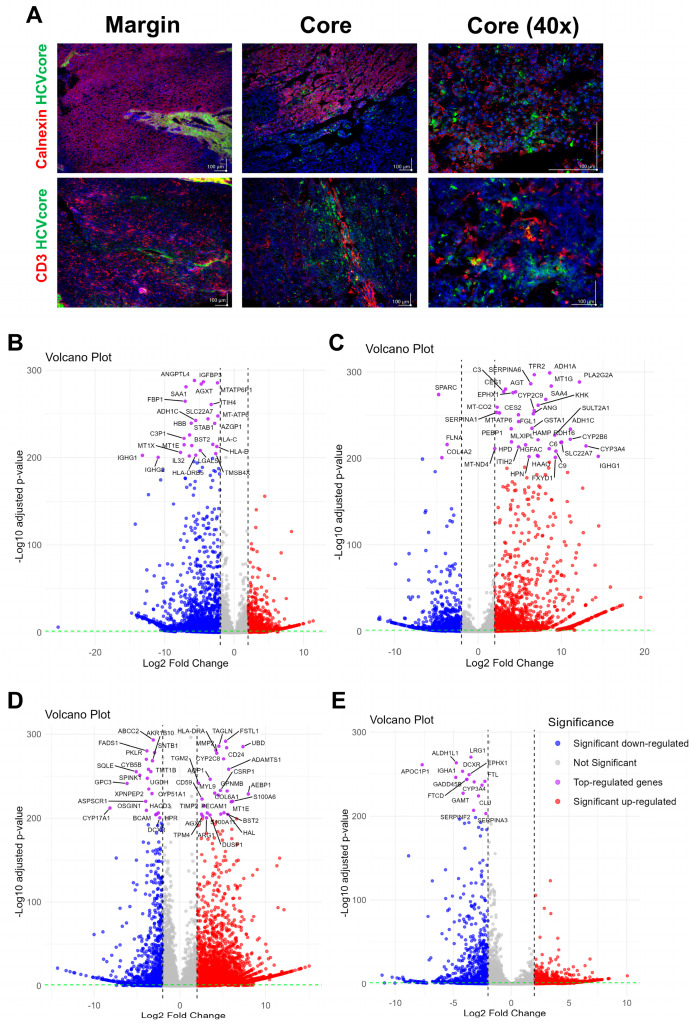
Comparative HCVcore protein occurrence and RNASeq-based analysis of tumor regions and conditions. (**A**) presents immunofluorescence staining of HCC tumor tissue, comparing the margin and core regions. The top row shows the localization of Calnexin (red), HCV core (green), and their overlay in both the tumor margin and core. The middle row displays staining for CD3 (red) and HCV core (green), highlighting the immune cell infiltration in relation to viral presence within the tumor. Scale bars represent 100 µm distance. (**B**–**E**) present volcano plots of differential gene expression analysis calculated as follows: (**B**) tumor core tissue against margin tissue, (**C**) tumor core isolate against the margin tissue, (**D**) tumor core tissue against healthy hepatocytes, and € tumor margin tissue against healthy hepatocytes. The x-axis represents the log2 fold change in gene expression, while the y-axis shows the −log10 adjusted *p*-value, indicating the statistical significance of these changes. Genes significantly downregulated are highlighted in blue, those significantly upregulated in red, and non-significant changes are in gray. The top-regulated genes (*p*-value < 200) are labeled and represented in purple.

**Figure 3 viruses-17-00653-f003:**
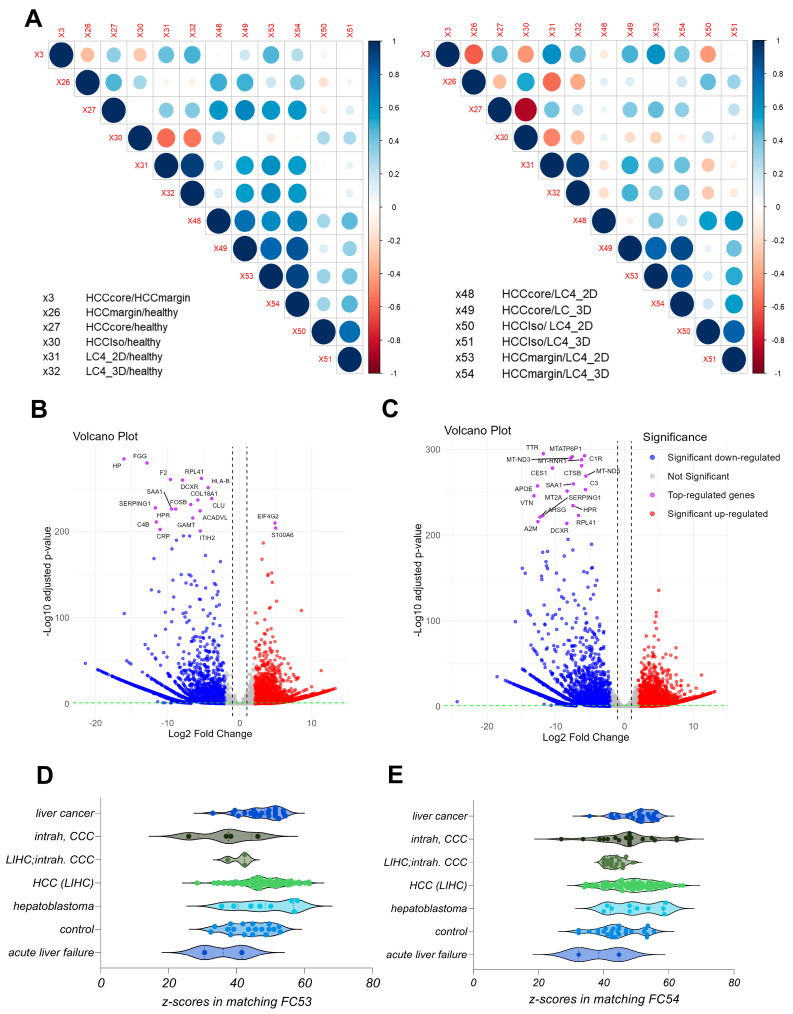
Correlation analysis of patient-derived specimens and DEG presentation of LC4 cells against tumor regions. (**A**) represents a correlation matrix displaying pairwise Pearson correlations between datasets of calculated z-scores of canonical pathways (**right panel**) and toxicity pathways (**left panel**) for the indicated DEG comparisons. Blue and red circles indicate the strength and direction of the correlation (blue = positive; red = negative). The size of the circle corresponds to the magnitude of the correlation coefficient, ranging from −1 to 1. (**B**,**C**) represent volcano plots of DEG comparisons between generated cell line LC4 in 2D (**B**) and 3D (**C**) conditions against the margin tissue. The volcano plots illustrate the log2 fold change in gene expression (x-axis) versus the −log10 adjusted *p*-value (y-axis). Each point represents a gene, with the x-axis showing the log2 fold change in expression levels between the margin and centrum and the y-axis showing the −log10 adjusted *p*-value. Blue points indicate significantly downregulated genes (adjusted *p*-value < 0.05), red points indicate upregulated genes, and purple points highlight the top-regulated genes based on fold change (*p*-value <200). Grey points represent genes with non-significant changes, defined by adjusted *p*-value > 0.05 and log2 fold change between 2, −2. In (**D**,**E**), the highest matches, indicated by high z-scores, from the analysis match function in IPA were visualized form the different disease subtypes. In (**D**), we compare the analysis between LC4_2D cells and the margin tissue (FC53) and in (**E**), we show the global comparison match with the 3D condition (FC54) compared to margin tissue from the origin. Every dot represents one single analysis of the global dataset from IPA. For matching, the analysis was reduced to liver, human, and the overall z-scores higher or lower than 10. In the graphs (**D**,**E**), all highest similarities were plotted and group as in IPA.

**Figure 4 viruses-17-00653-f004:**
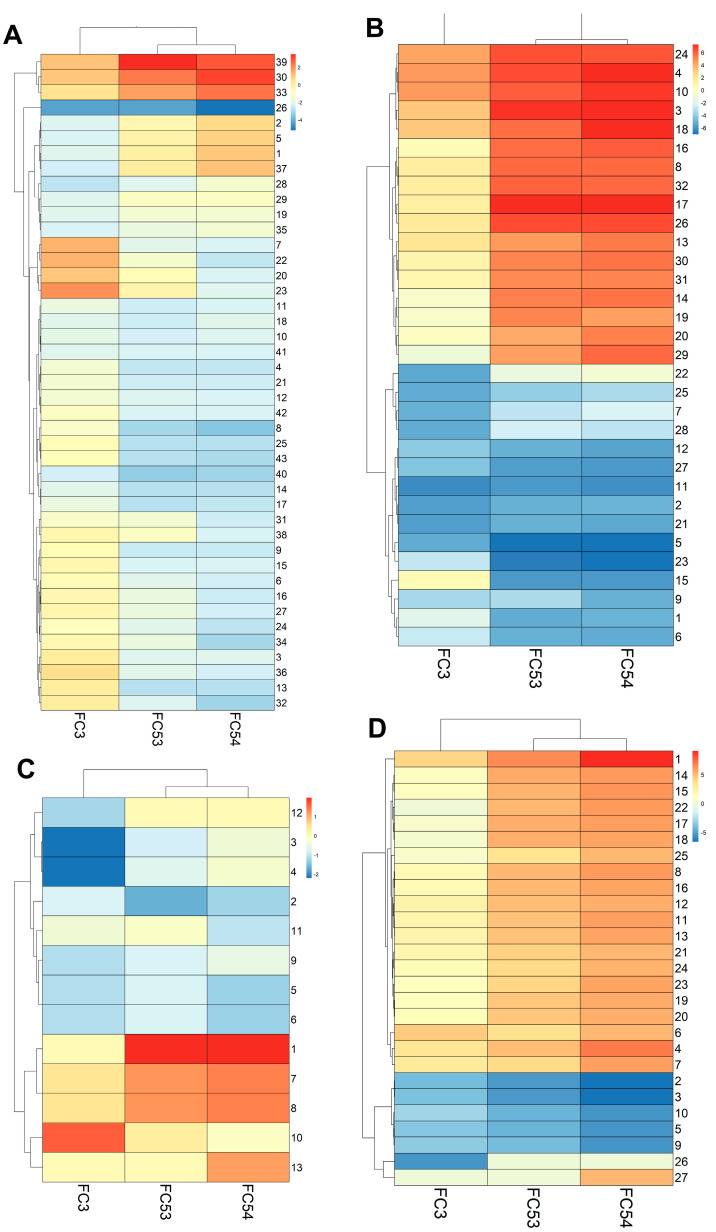
Comparison of relevant canonical pathways, disease and biofunctions, and toxicity functions in patient-derived specimens. In Figure 4, pathway comparison was performed from DEG analysis, calculated against the tumor margin expression for the experimental conditions: core tumor tissue (Analysis 3), LC4 cells in 2D culture (Analysis 53), and LC4 cells in 3D culture (Analysis 54). Z-scores > −5 and < 5 with log_10_*p*-values > 1.3 have been compared. The heatmaps show the z-scores of pathway activation, with red indicating upregulation and blue indicating downregulation. (**A**) metabolic canonical pathways and (**B**) signaling canonical pathways are shown for the three comparisons. In (**C**), the toxicity functions and (**D**), diseases and biofunctions were compared between the generated cell line LC4 and the core region of the parental tumor. For comparative illustration pathways were numbered using pheatmap package in Rstudio and decoded in Table 1, Table 2, Table 3 and Table 4.

**Figure 5 viruses-17-00653-f005:**
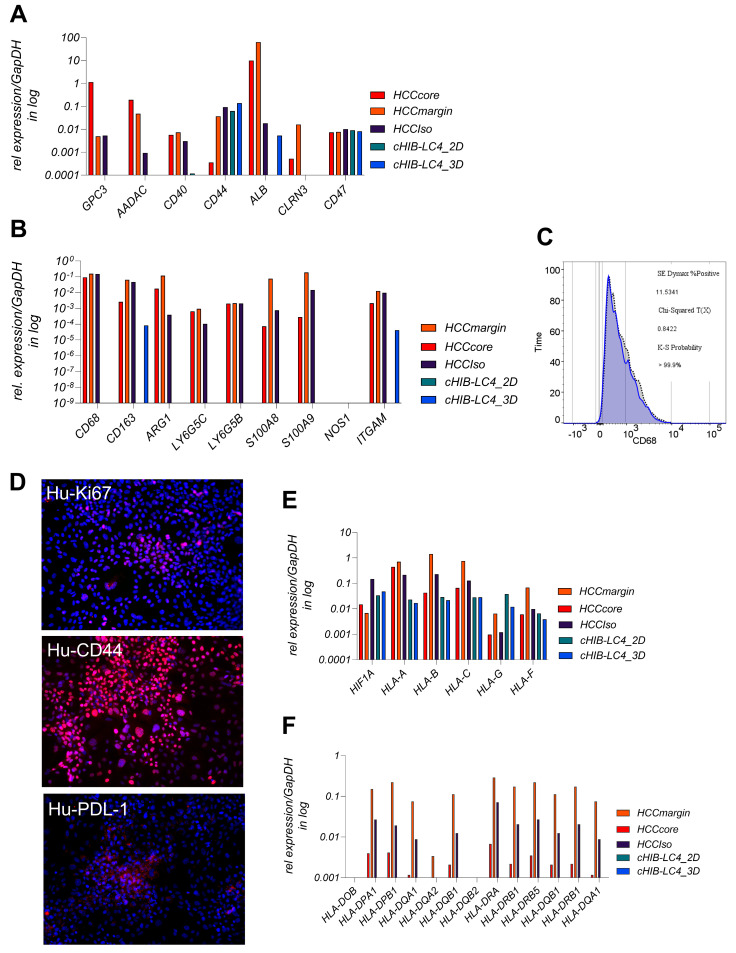
Liver-tumor- and immune-recognition-related gene and protein expression profiles of patient-derived LC4 cells. (**A**,**B**,**E**,**F**) display mRNA expression levels of selected genes in different cell fractions: HCC core, margin tissue, HCCIso (freshly isolated tumor cells from the core region), LC4_2D (generated cell line in 2D culture conditions), and LC4_3D (generated cell line in 3D culture conditions), normalized against GapDH expression and plotted on log10 scale. In (**A**), liver and tumor specific genes including GPC3, CD40, CD44, CD47, AADAC, ALB, and CLRN3 were analyzed in the indicated fractions. (**B**) shows the mRNA expression level of genes associated with tumor-associated macrophages (TAMs) and myeloid-derived suppressor cells (MDSCs) in the same samples. The genes analyzed include CD68, ARG1, CD163, ITGAM, LY6G5B, LY6G5C, NOS1, S100A8, and S100A9. This comparison highlights the differences in the expression of immune-related genes under different experimental conditions, showing changes in the tumor microenvironment during cell culture and generating the LC4 cell lines. (**C**) presents flow cytometry analysis for CD68, a marker for macrophages, in LC4 cells. The plot displays the percentage of CD68-positive cells, along with statistical parameters such as the K-S probability, Chi-Squared value, and SE Dynmax %Positive. (**D**) shows immunofluorescence staining of LC4 cells in 2D culture conditions for Ki67, PDL-1, and CD44. The images display the expression of Ki67 (red), PDL-1 (red), and CD44 (red), with nuclei counterstained with DAPI (blue). These staining patterns illustrate cell proliferation (Ki67), immune checkpoint expression (PDL-1), and stemness markers (CD44), offering the visual confirmation of the molecular profiles. In (**E**,**F**), the differential expression pattern of genes related to HIF1A, MHC class I, and MHC class II were shown detected in the patient-derived tissue regions and cells compared to the generated LC4 cell line.

**Figure 6 viruses-17-00653-f006:**
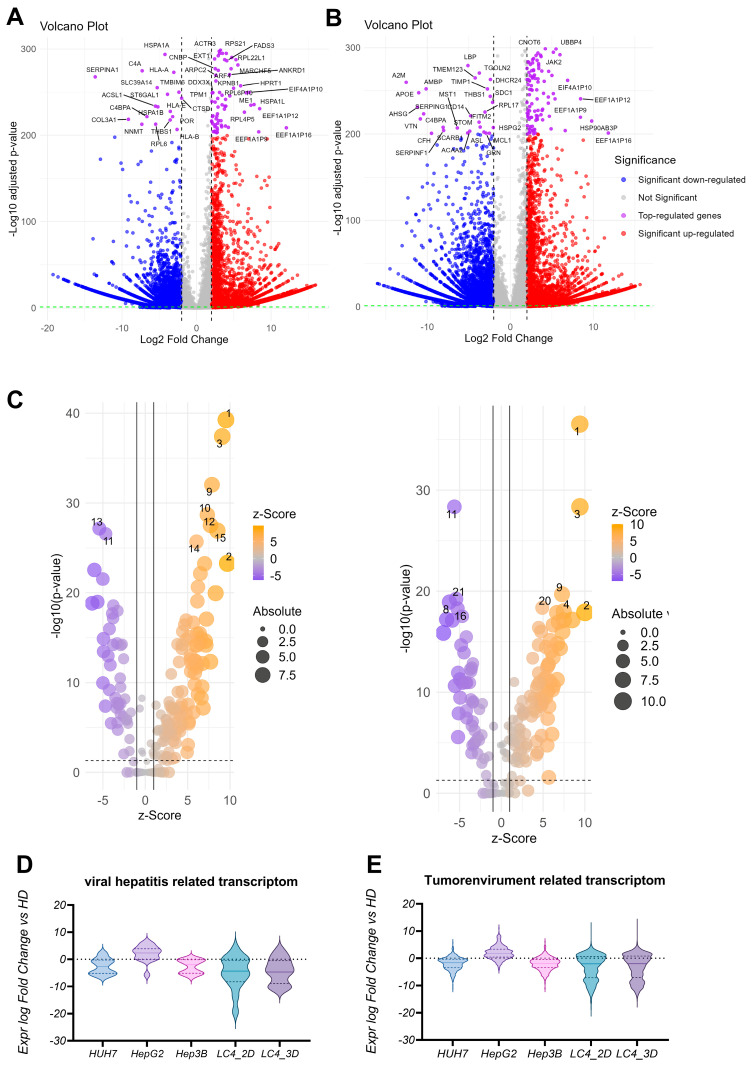
DEG and pathway activation analysis of LC4 cells compared to healthy donors and immortalized cell lines. (**A**,**B**) showing the differential gene expression between LC4 cells cultured under 2D (LC4_2D; **A**) and 3D (LC4_3D; **B**) conditions compared with the expression counts detected in RNASeq of healthy hepatocytes. The volcano plots illustrate the log2 fold change in gene expression (x-axis) versus the −log10 adjusted *p*-value (y-axis). Each point represents a gene, with the x-axis showing the log2 fold change in expression levels and the y-axis showing the −log10 adjusted *p*-value. Blue points indicate significantly downregulated genes (adjusted *p*-value < 0.05), red points indicate upregulated considerably genes, and purple points highlight the top-regulated genes based on fold change, with *p*-values < 200. Grey points represent genes with non-significant changes, defined by adjusted *p*-value > 0.05 and log2 fold change between 2, −2. In (**C**), the DEG analysis from (**A**,**B**) were used for predictive canonical pathway comparison. The x-axis shows the z-score, indicating the direction and magnitude of expression change, while the y-axis displays the −log10 (*p*-value), representing the statistical significance of the changes. The size of the dots corresponds to the absolute expression levels, and the color gradient indicates the z-score, with orange representing upregulation and purple representing downregulation. Most prominent pathways were numbered and listed in Table 5. In (**D**,**E**), the expression of genes involved in the pathways: viral hepatitis (**D**) and tumor environment (**E**) were analyzed in fold change against the healthy hepatocytes for commonly used immortalized cell lines Huh7, Hep3B, and HepG2 in comparison to generated LC4 cells in 2D and 3D conditions.

**Figure 7 viruses-17-00653-f007:**
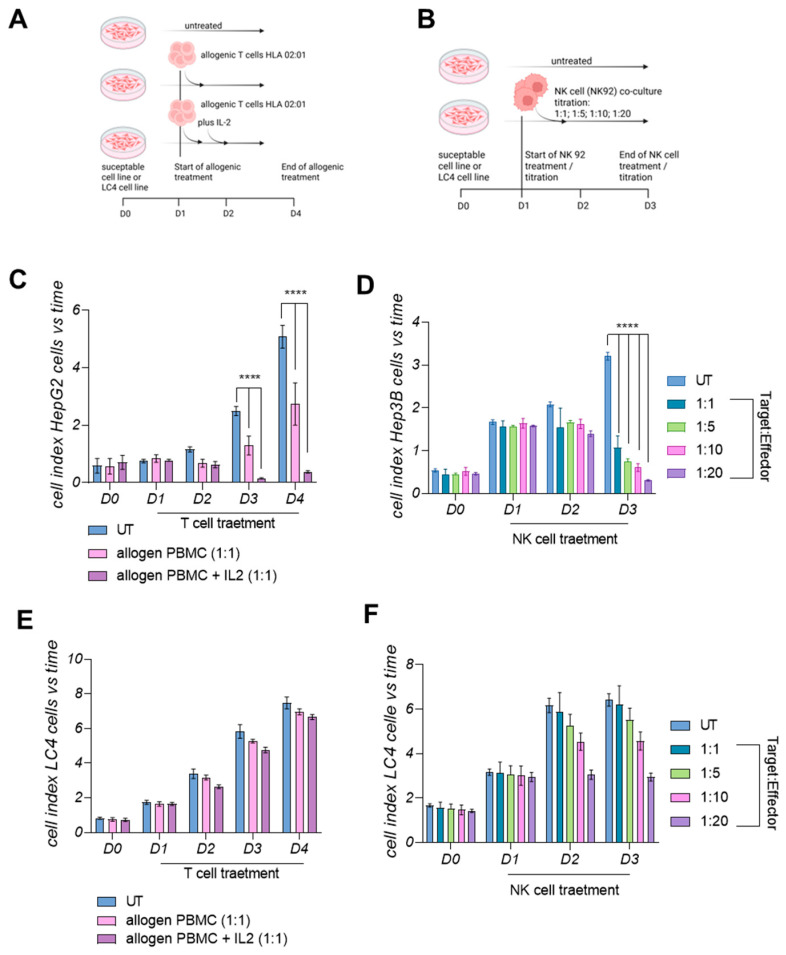
Allogenic immune cell transfer in vitro using LC4 cells and susceptible immortalized HCC cell lines. In (**A**,**B**), the experimental settings are displayed. HCC cells and LC4 cells were seeded in 2D culture. (**A**) The adoptive Transfer of T cells and (**B**) the transfer of active NK92 cells. (**C**–**F**) The Xcelligence measurement with the in (**A**,**B**) displayed settings. The allogenic T cell transfer to (**C**) HepG2.1.3 and (**E**) LC4 cells are also shown (n = 5 replicates; 3 groups), while (**D**,**F**) display Hep3B and LC4 cells, respectively, after NK92 transfer (n = 2 × 3 technical replicates; 5 groups). Xcelligence measurement started after seeding on D0 until the end of experiment D4 for T cells or D3 for NK cells. Statistical outputs are indicated in the figure legends; *p*-values were plotted in the graph **** *p* ≤ 0.0001.

**Figure 8 viruses-17-00653-f008:**
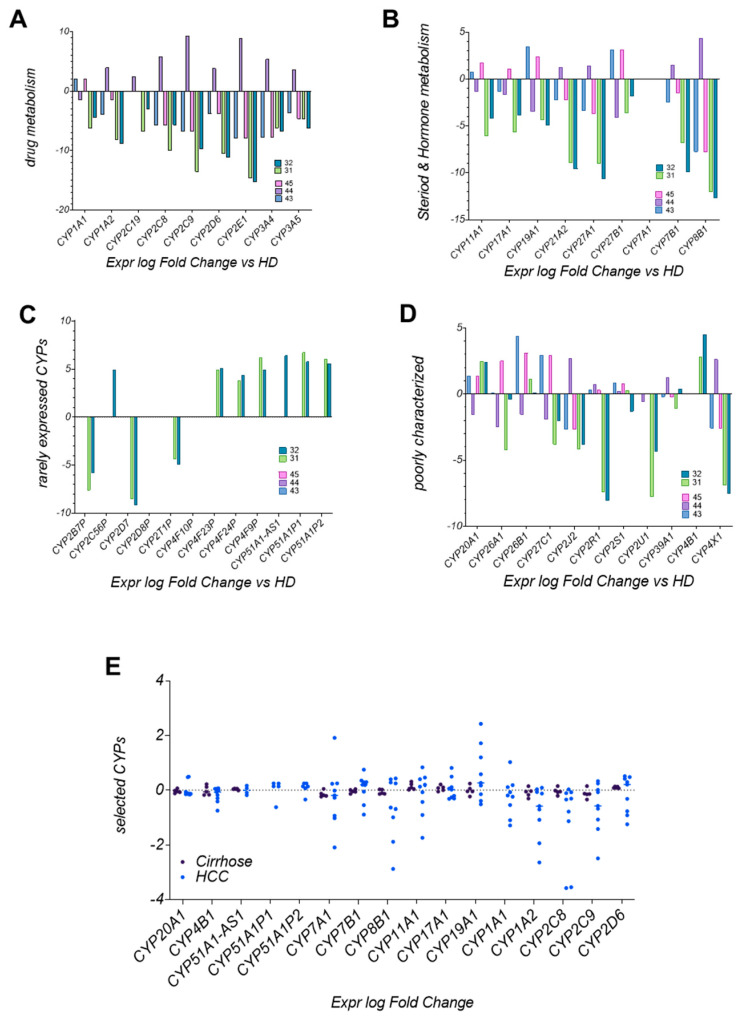
DEG of cytochrome P450 enzymes in patient-derived cells, immortalized cell lines, and global datasets. In (**A**,**B**), bar plots visualize the log fold change analysis of CYP enzymes involved in drug metabolism (**A**) and steroid and hormone metabolism (**B**). Rarely expressed and poorly characterized CYPs were displayed in (**C**,**D**). The expression profiles were analyzed using healthy donor samples for normalization. The indicated CYP enzymes were analyzed for LC4 cells in 2D culture (31) and LC4 cells in 3D culture (32), with the immortalized cell lines HUH7 = 43, HepG2 = 44, and Hep3B = 45. In (**E**), a dot plot was used for comparing the expression log fold change of indicated CYP enzymes in a global IPA provided dataset of patients with untreated cirrhosis (n = 5) and untreated diagnosed HCC (n = 9).

**Figure 9 viruses-17-00653-f009:**
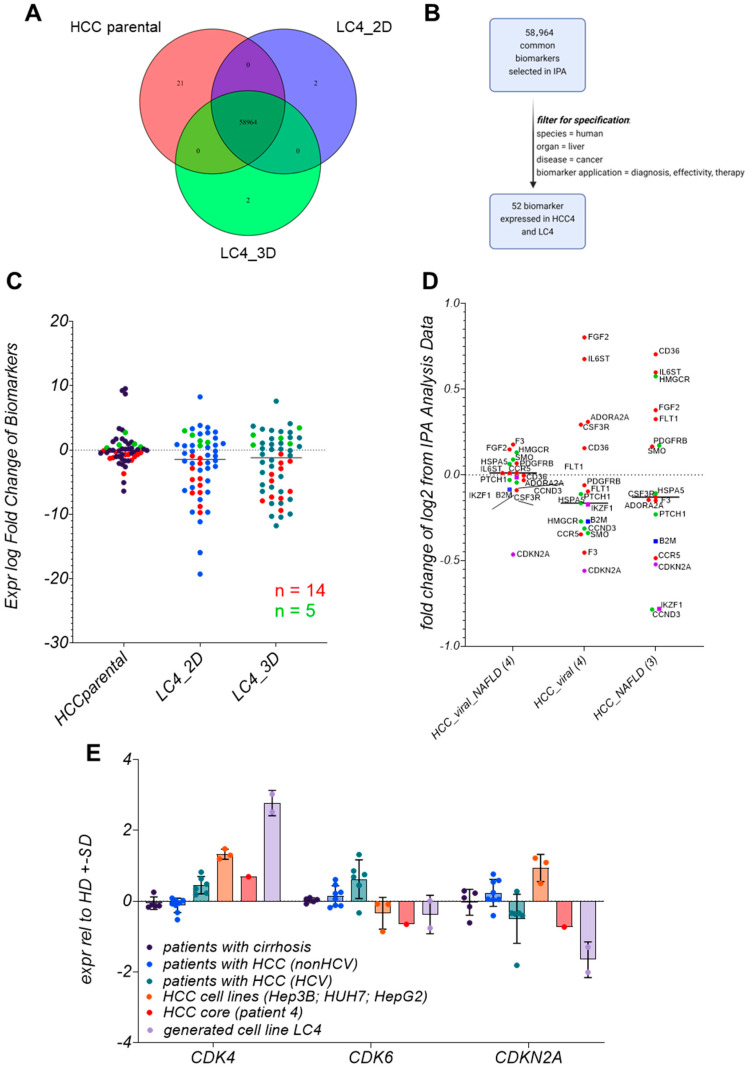
Identification and classification of disease-related consistently expressed biomarkers presented in LC4 cells. In (**A**), the Venn diagram illustrates the overlap of biomarkers identified in three experimental conditions: HCC core tumor tissue (HCC core), LC4 cells in 2D culture (LC4_2D), and LC4 cells in 3D culture (LC4_3D). The diagram shows the number of unique and shared biomarkers across these conditions, with 58,964 overlapping biomarkers. In (**B**), the filter progress within the ingenuity pathway analysis was visualized schematically. The dataset was filtered for “human liver cancer” biomarkers relevant to diagnosis, effectiveness, and therapy, resulting in 52 biomarkers expressed in both parental tumor cells and LC4 cells. In (**C**), the scatter dot plot displays the expression log fold change of the selected 52 biomarkers in the three experimental conditions. Each dot represents a biomarker, and the distribution illustrates the consistent regulation among the different conditions. Green dots indicate consistently upregulated biomarkers (n = 5) and red dots consistently downregulated biomarkers (n = 14). In (**D**), the dot plot analysis shows the fold change of consistently regulated biomarkers derived from cells from patient 4 in log2, based on a global IPA analysis dataset comparing relevant specific conditions in HCC patients, by displaying the median of 4 viral/NAFLD patients, 4 viral HCC patients, and 3 NAFLD patients. Red dots present downregulated while green dots represent upregulated biomarkers detected in LC4 cells. Purple and blue squares as well as purple dots identify biomarkers consistently regulated in all three conditions of HCCs. In the bar plot displayed in (**E**) the expression log fold change of CDK4, CDK6, and CDKN2A across different patient groups and cell lines compared to healthy donors. The analysis includes samples from patients with cirrhosis (n = 5), patients with HCC (non-HCV; n = 8), patients with HCC (HCV; n = 6,), HCC cell lines (Hep3B and HUH7), the patient 4 isolate (HCC4Iso n = 1), and the generated cell line LC4 (n 02). The standard deviation is represented by the error bars of each column.

**Table 1 viruses-17-00653-t001:** List of metabolic canonical pathways.

Number	Pathway
1	3-phosphoinositide Biosynthesis
2	3-phosphoinositide Degradation
3	Acetate Conversion to Acetyl-CoA
4	Bile Acid Biosynthesis, Neutral Pathway
5	D-myo-inositol-5-phosphate Metabolism
6	Dopamine Degradation
7	Estrogen Biosynthesis
8	Ethanol Degradation II
9	Ethanol Degradation IV
10	Fatty Acid β-oxidation I
11	Glutaryl-CoA Degradation
12	Glutathione Redox Reactions I
13	Glutathione-mediated Detoxification
14	Glycine Betaine Degradation
15	Histamine Degradation
16	Histidine Degradation VI
17	Isoleucine Degradation I
18	Ketogenesis
19	Ketolytic
20	Melatonin Degradation I
21	NAD biosynthesis II (from tryptophan)
22	Nicotine Degradation II
23	Nicotine Degradation III
24	Noradrenaline and Adrenaline Degradation
25	Oxidative Ethanol Degradation III
26	Oxidative Phosphorylation
27	Phenylalanine Degradation IV (Mammalian, via Side Chain)
28	Phospholipases
29	Proteinoid Biosynthesis
30	Pyridoxal 5’-phosphate Salvage Pathway
31	Retinoate Biosynthesis I
32	Retinol Biosynthesis
33	Salvage Pathways of Pyrimidine Ribonucleotides
34	Serotonin Degradation
35	Sucrose Degradation V (Mammalian)
36	Super pathway of Cholesterol Biosynthesis
37	Super pathway of Inositol Phosphate Compounds
38	The Visual Cycle
39	tRNA Charging
40	Tryptophan Degradation III (Eukaryotic)
41	Tyrosine Degradation I
42	Urea Cycle
43	Valine Degradation I

**Table 2 viruses-17-00653-t002:** List of signal canonical pathways.

Number	Pathway
1	Aspirin ADME
2	Atherosclerosis Signaling
3	Cell Cycle Checkpoints
4	Chromatin organization
5	Complement cascade
6	Complex I biogenesis
7	Dendritic Cell Maturation
8	Deubiquitinating
9	Eukaryotic Translation Elongation
10	Histone Modification Signaling Pathway
11	Immunoregulatory interactions between a Lymphoid and a non-Lymphoid cell
12	Interferon alpha/beta signaling
13	Intra-Golgi and retrograde Golgi-to-ER traffic
14	L1CAM interactions
15	Mitochondrial translation
16	Mitotic G2-G2/M phases
17	Mitotic Metaphase and Anaphase
18	Mitotic Prometaphase
19	Neddylation
20	Nuclear Cytoskeleton Signaling Pathway
21	Pathogen Induced Cytokine Storm Signaling Pathway
22	Phagosome Formation
23	Phase I—Functionalization of compounds
24	Processing of Capped Intron-Containing Pre-mRNA
25	Production of Nitric Oxide and Reactive Oxygen Species in Macrophages
26	Protein Ubiquitination Pathway
27	Respiratory electron transport
28	Response to elevated platelet cytosolic Ca2+
29	RHO GTPase cycle
30	RHO GTPases Activate Forming
31	Signaling by NTRK1 (TRKA)
32	TCF dependent signaling in response to WNT

**Table 3 viruses-17-00653-t003:** List of toxic pathways.

Number	Pathway
1	Infection by hepatitis B virus
2	Apoptosis of liver cell lines
3	Proliferation of liver cells
4	Proliferation of hepatic stellate cells
5	Cell death of liver cells
6	Necrosis of liver
7	Inflammation of liver
8	Viral hepatitis
9	Proliferation of hepatocytes
10	Liver tumor
11	Increased Levels of LDH
12	Cell death of liver cell lines
13	Fibrosis of liver

**Table 4 viruses-17-00653-t004:** List of diseases and biofunction pathways.

Number	Pathway
1	Cell proliferation of tumor cell lines
2	Sensitivity of tumor cell lines
3	Sensitivity of cells
4	Cell proliferation of carcinoma cell lines
5	Cell death of tumor cell lines
6	Cell proliferation of hepatoma cell lines
7	Cell proliferation of adenocarcinoma cell lines
8	Infection by RNA virus
9	Necrosis
10	Apoptosis of tumor cell lines
11	Migration of carcinoma cell lines
12	Infection of tumor cell lines
13	Cell movement of carcinoma cell lines
14	Cell survival
15	Viral Infection
16	Infection of cells
17	Cell viability
18	Cell viability of tumor cell lines
19	Infection by Reoviridae
20	HIV infection
21	Cell proliferation of colorectal cancer cell lines
22	Infection by HIV-1
23	Migration of tumor cell lines
24	Cell proliferation of liver cancer cell lines
25	Cell movement of tumor cell lines
26	Mobilization of Ca2+
27	Viral life cycle

**Table 5 viruses-17-00653-t005:** List of descriptive encoding of canonical pathways analyzed in IPA with high z-score alterations and *p*-values in comparative analysis.

Number	Pathway
1	Neutrophil degranulation
2	Response to elevated platelet cytosolic Ca2+
3	EIF2 Signaling
4	LXR/RXR Activation
5	Acute Phase Response Signaling
6	Post-translational protein phosphorylation
7	Regulation of Insulin-like Growth Factor (IGF) transport and uptake by IGFBPs
8	DHCR24 Signaling Pathway
9	Eukaryotic Translation Initiation
10	Seleno amino acid metabolism
11	Protein Ubiquitination Pathway
12	Response of EIF2AK4 (GCN2) to amino acid deficiency
13	Eukaryotic Translation Elongation
14	Eukaryotic Translation Termination
15	SRP-dependent co-translational protein targeting to membrane
16	RHO GTPase cycle
17	Nonsense-Mediated Decay (NMD)
18	Pulmonary Fibrosis Idiopathic Signaling Pathway
19	Production of Nitric Oxide and Reactive Oxygen Species in Macrophages
20	Mitotic Metaphase and Anaphase
21	Class I MHC mediated antigen processing and presentation
22	Extracellular matrix organization
23	Integrin cell surface interactions
26	PTEN Regulation
31	Nuclear Cytoskeleton Signaling Pathway
32	Degradation of beta-catenin by the destruction complex
34	Mitotic G2-G2/M phases
38	Deubiquitinating
44	Transcriptional regulation by RUNX3
53	Cell Cycle Checkpoints
58	Neddylation
70	Complement cascade
89	Pathogen Induced Cytokine Storm Signaling Pathway
110	Cell surface interactions at the vascular wall
123	Binding and Uptake of Ligands by Scavenger Receptors
141	Immunoregulatory interactions between a Lymphoid and a non-Lymphoid cell
148	Fcgamma receptor (FCGR) dependent phagocytosis

## Data Availability

The RNASeq datasets are available in the global datasets in the IPA SW form Qiagen. All other datasets are available by a direct request to the first author.

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
