# Peer review of "Classification of the LC4 Primarily-like Cell Line—Recapitulating a CDK4 Overexpressing Immune Evasive HIV-HCV-Induced HCC"

_viruses, 2025, doi:10.3390/v17050653_

Round 1

Reviewer 1 Report

Comments and Suggestions for Authors

This manuscript described the development of a new cell line, LC4, from the core region of hepatocellular carcinoma (HCC) with underlying HIV-HCV co-infection. The new primary-like cell line has the immunosuppressive and metabolic characteristics of the parental cancer region such as down-regulation of drug-metabolizing CYP enzymes. This new cell line may be useful as a model of studying the mechanisms of hepatocellular carcinoma development and for pre-clinical testing of potential drugs for treatment of HCC. This manuscript is well written and I have no comments or questions on any aspect of this interesting and informative study presented in this manuscript.

Author Response

We thank the reviewer for the very positive assessment of our work.

Reviewer 2 Report

Comments and Suggestions for Authors

The manuscript titled "Classification of the LC4 primarily-like cell line - recapitulating a CDK4 overexpressing immunosuppressive HIV-HCV-induced HCC" presents an interesting study on the classification and characterization of a novel HCC cell line derived from a patient co-infected with HIV and HCV. The study aims to comprehensively analyze the LC4 cell line, which could serve as a valuable model for preclinical testing in HCC research. However, the manuscript in its current form lacks crucial details, particularly in the Materials and Methods section, which hampers the reproducibility and clarity of the study.

The abstract concisely summarizes the study's objectives, methods, and findings. However, it would benefit from a more explicit statement of the study's significance and potential impact on HCC research.

The introduction effectively sets the context for the study by discussing the prevalence and challenges associated with HCC, particularly in the context of HIV-HCV co-infection. The rationale for developing the LC4 cell line is well-articulated.

This section is the most concerning part of the manuscript. Several key details are missing, which are essential for the study's reproducibility.

For the Human Sample Collection, since the ethical approval and source of the samples are mentioned, detailed information on the sample collection process, inclusion/exclusion criteria, and demographic characteristics of the donors should be provided.

What specific demographic characteristics of the donors were involved in the human sample collection? 

What unique advantages does the LC4 cell line offer compared to existing HCC cell lines and models? 

What future research directions could be pursued to address the limitations highlighted in the study?

The method for HLA typing is described, but the specific reagents, kits, and protocols used should be clearly stated. The conditions for cell culture are briefly mentioned, but detailed protocols, including media composition, passaging techniques, and any specific treatments or conditions, should be included. The RNA isolation and sequencing methods are outlined, but more specific details on the library preparation, sequencing parameters, and quality control measures are needed.

Data Processing and Analysis pipelines used for data processing and analysis are mentioned, but the specific tools, versions, and parameters used for each step should be clearly described. This includes details on the differential expression analysis, pathway analysis, and any statistical methods employed.

The results section is comprehensive and presents a detailed analysis of the LC4 cell line. However, some figures and tables are not sufficiently described in the text. Each figure and table should have a clear and concise legend explaining the data and key findings.

Discussion of the implications of the findings in the context of existing literature and their potential impact on future research could strengthen the interpretation of the results.

The discussion section provides a good overview of the study's findings and their significance. However, it would benefit from a more in-depth discussion of the study's limitations and potential future directions for research.

The discussion should also address how the LC4 cell line compares to existing HCC cell lines and models and what unique advantages it offers for HCC research.

The conclusion is well-written and summarizes the study's key findings. However, it could be strengthened by highlighting the study's implications for clinical practice and future research.

The references are relevant and up-to-date. However, some of the citations in the text do not correspond to the references listed at the end of the manuscript. This should be corrected for accuracy. The manuscript requires significant revisions, particularly in the Materials and Methods section, to provide the necessary details for reproducibility and clarity. The authors should ensure that all figures and tables are sufficiently described in the text and that the legends are clear and concise. The discussion section should include a more in-depth analysis of the study's limitations and future directions.

Author Response

Comments and Suggestions for Authors

The manuscript titled "Classification of the LC4 primarily-like cell line - recapitulating a CDK4 overexpressing immunosuppressive HIV-HCV-induced HCC" presents an interesting study on the classification and characterization of a novel HCC cell line derived from a patient co-infected with HIV and HCV. The study aims to comprehensively analyze the LC4 cell line, which could serve as a valuable model for preclinical testing in HCC research. However, the manuscript in its current form lacks crucial details, particularly in the Materials and Methods section, which hampers the reproducibility and clarity of the study.

Abstract:

The abstract concisely summarizes the study's objectives, methods, and findings. However, it would benefit from a more explicit statement of the study's significance and potential impact on HCC research.

We thank the reviewer for the comment. We improved the conclusion statement: “We classified LC4 cells as an individual immunosuppressive and highly progressive primary-like HCC cell line. LC4 cells are applicable as a model for pre-clinical drug testing, minimizing the lack of pre-clinical models in HCV-HIV-induced HCC research. “ in the manuscript to point out the significance of the study on HCC research. (line 30-33)

The introduction effectively sets the context for the study by discussing the prevalence and challenges associated with HCC, particularly in the context of HIV-HCV co-infection. The rationale for developing the LC4 cell line is well-articulated.

Methods:

This section is the most concerning part of the manuscript. Several key details are missing, which are essential for the study's reproducibility.

We thank the reviewer for requesting detailed information in the M&M section. We like to point out, that we included additional information in the supplementary methods section. And we highlighted the subunits of the M&M section better.

For the Human Sample Collection, since the ethical approval and source of the samples are mentioned, detailed information on the sample collection process, inclusion/exclusion criteria, and demographic characteristics of the donors should be provided. What specific demographic characteristics of the donors were involved in the human sample collection?

We thank the reviewer for the request on the sample collection more in detail. As stated in the manuscript, we previously described the generation of the primary-like cell lines in MDPI Cells. However, we included following information’s in the current manuscript: “Samples from male and female patients suffering from HCC between 18 – 99 years were collected with no demographic exclusion characteristics. ” (Line 87 – 89).

What unique advantages does the LC4 cell line offer compared to existing HCC cell lines and models?

We like to clarify: The patient 4 suffered from a highly progressive HCC with has been developed on the viral co-infection HCV-HIV. In HCC research model reflecting this entity are rare. Therefore, the LC4 patient-derived model gave the opportunity for pre-clinical drug testing in a clinically relevant setting.

What future research directions could be pursued to address the limitations highlighted in the study?

We thank for the question: The limitation of the generated cell line lines in the adaptation of cellular processes and the proliferation to the cell culture conditions. However, this limitation is a cell culture artefact and can be reduced or overcome by the usage of LC4 cell in vivo. As mentioned in the manuscript, the cell line was generated by using immunosuppressive mice and performing orthotopic transplantation. To overcome the cell culture limitations, the physiological environment will be an alternative. Therefore, the next step would be to analyse the transplanted mice in regards of the adoption and changes to both, the cell culture and the parental conditions.

The method for HLA typing is described, but the specific reagents, kits, and protocols used should be clearly stated.

We thank for the comment, but we need to clarify the request. We described the standard diagnostic method including all kits and protocols in line 99 – 112. What exact is the missing information?

The conditions for cell culture are briefly mentioned, but detailed protocols, including media composition, passaging techniques, and any specific treatments or conditions, should be included.

We thank for the comment. We like to point out, that we described a detailed protocol in MPDI Cells Methods for the generated cell lines previously and the immortalized cell lines as mentioned in the cell culture section. The same conditions were used in the underlying study.  For the LC4 Cell line we included the Data sheet in the supplementary method section, to specify the media and characteristics. We stated in the M&M section in line 114: “For detailed information on LC4 cells a data sheet is available in the supplementary section. “

 The RNA isolation and sequencing methods are outlined, but more specific details on the library preparation, sequencing parameters, and quality control measures are needed.

We thank for the comment and extend the description in the manuscript, in line 162, 166-167, 173, 174-175:

RNA Isolation and Sequencing. Total RNA was extracted from primary hepatocyte samples, the primary HCC cell line LC4, and immortalized HCC cell lines (HepG2, Huh7, and Hep3B) using the RNeasy Mini Kit (Qiagen), following the manufacturer's protocol. The quality and quantity of RNA were assessed using a NanoDrop spectrophotometer (ThermoFisher Scientific) and an Agilent Bioanalyzer 2100 (Agilent Technologies), using the High Sensitivity DNA Chips (Cat: 5067-4626). High-quality RNA samples with an RNA Integrity Number (RIN) greater than 7.0 were selected for sequencing.

Library Preparation and RNA Sequencing. RNA sequencing libraries were pre-pared using the TruSeq Stranded mRNA Library Prep Kit and the index adapter kit IDT-Limn RNA UD Indexes set a Ligation according to the manufacturer's instructions (Illumina). MRNA was purified from total RNA using poly-T oligo-attached magnetic beads and fragmented into small pieces. First-strand cDNA was synthesized using random hexamer primers and reverse transcriptase. This was followed by second-strand cDNA synthesis, end repair, A-tailing, adapter ligation, and PCR amplification to enrich the cDNA fragments. The libraries were quantified using a Qubit fluorometer (Ther-moFisher Scientific), using the Qubit 1X dsDNA HS Assay Kit (Q33230) and assessed for size distribution using the Agilent Bioanalyzer 2100. The libraries were then sequenced on the Illumina NextSeq 1000/2000 platform, generating 200 bp paired end reads. Tech-nical replicates were used for RNAseq for each sample (n = 2).

Data Processing and Analysis pipelines used for data processing and analysis are mentioned, but the specific tools, versions, and parameters used for each step should be clearly described. This includes details on the differential expression analysis, pathway analysis, and any statistical methods employed.

We thank for the comment. We announced the specific tools and parameters in the supplementary M&M section. Additionally, we included the versions to the R-studio packages used to perform the analysis. We hope that the additional information clarify the missing points. We stated in line 184: “Detailed information was described in the supplementary methods. “ In the supplementary section, changes were made visible by underlining.

The results section is comprehensive and presents a detailed analysis of the LC4 cell line. However, some figures and tables are not sufficiently described in the text. Each figure and table should have a clear and concise legend explaining the data and key findings.

We thank for the comment. We did include a description for every table and figure. However, we could not exactly see where there was a missing information. It would be very helpful if the Reviewer could please state a clear insufficient description of a Table or Figure.

Discussion of the implications of the findings in the context of existing literature and their potential impact on future research could strengthen the interpretation of the results.

The discussion section provides a good overview of the study's findings and their significance. However, it would benefit from a more in-depth discussion of the study's limitations and potential future directions for research.

We thank the reviewer and see the point of pointing out the limitations of the LC4 cell line. However, we stated that the 3D culture was found to be closer in the correlation and pathway analysis to the parental tissue. In this case, the 2D condition show the limitation in reflecting the parental pathway composition but reflecting the immunosuppressive nature of the parental tumour. Therefore, both conditions can be used for drug testing.

The discussion should also address how the LC4 cell line compares to existing HCC cell lines and models and what unique advantages it offers for HCC research.

We thank the reviewer for making that point. In this study we compared on the one hand the parental and healthy liver cells and on the other hand immortalized HCC cell lines. The focus in comparing commonly used HCC cell lines to LC4 cells was to understand the metabolic characteristics and expression of CYP genes. A well-known limitations of commonly used HCC cell lines is the insufficient refection of the clinical metabolic status in vitro. The aim was to categorize the LC4 cells as 2D and 3D conditions in the landscape of 3 know HCC cell lines. We found that HUH7 and Hep3B cell lines reflect the low LC4 CYP expression for drug metabolism related genes, while HepG2 cells were mainly upregulated. LC4 cells were found to express rarely expressed CYPs while none of the commonly used HCC cell lines expressed these CYP. In general, the expression of CYP genes was found to be inhibited or significant downregulated in LC4 cells due to the viral infection. The unique advantage of the LC4 cells is the usage as a model for chemotherapy resistance which is directly connected to the low CYP expression status. Therefore, the LC4 cell line has a unique advantage for HCC research by providing a low CYP expression model with a viral infection origin.  We discussed these points in the manuscript line 701-713.

The conclusion is well-written and summarizes the study's key findings. However, it could be strengthened by highlighting the study's implications for clinical practice and future research.

We thank the reviewer for the comment and concluded additionally: “ The low CYP expression and the immunosuppressive characteristic offers the possibility to perform drug screening assays in a clinically relevant setting for a better translation of pre-clinical results.” Line 799-801.

The references are relevant and up-to-date. However, some of the citations in the text do not correspond to the references listed at the end of the manuscript. This should be corrected for accuracy.

We thank the reviewer for critically reading the manuscript. We read truth the manuscript a the references and could not see the described discrepancy. Could the Reviewer please point out a specific reference or citation which is not citated correctly?

The manuscript requires significant revisions, particularly in the Materials and Methods section, to provide the necessary details for reproducibility and clarity. The authors should ensure that all figures and tables are sufficiently described in the text and that the legends are clear and concise. The discussion section should include a more in-depth analysis of the study's limitations and future directions.

In Summary, we Thank the reviewer for the comments and questions. We additionally discussed the open points and provide more information in the M&M section. However, some comments we will be thankful for a clarification from the reviewer’s side.

Reviewer 3 Report

Comments and Suggestions for Authors

This manuscript characterizes a cell line (LC4) derived from a patient suffering from HCC. 

The authors stated in the abstract: "We classified LC4 cells as an individual immunosuppressive and highly progressive primary-like HCC cell line, applicable as a model for pre-clinical testing"

This concept was reinforced in the introduction section last portion.

"Our key findings show that LC4 cells exhibited distinct characteristics reflect-ing the parental tissue, including the highly proliferative and immunosuppressive character displayed by the upregulation of genes in the PD-L1/VEGF pathway and the down-regulation of B2M, IKZF1, and CDKN2A, accompanied by CDK4 overexpression"

The analysis is well-performed and presented.  The main lack is that there is no experimental evidence that this cell line is indeed an immunosuppressive one or that it can be used in pre-clinical models.

The assertion: "Supplementary Figure 3B focuses on B2M, a key component of the MHC Class I complex crucial for antigen presentation. The downregulation of B2M in LC4 aligns with the results presented in Figure 5. The reduced antigen presentation will likely impair cytotoxic T-cell and natural killer cell responses. This downregulation is consistent with the immune-suppressive environment of the parental tissue, particularly in the context of PD-1/PD-L1 signaling"

It is well known that T cells can recognize the antigen in the context of HLA class I antigens. But it is also well known that the lack of HLA class I antigens can trigger NK cell activities. This would imply that the assertion regarding the supplementary figure and all the information on the immunosuppressive properties of this cell line are not clear and definitive evidence that this cell line is immunosuppressive.

The authors should analyze whether this cell line (compared with the others reported in the manuscript) is immunosuppressive. Does it not induce alloreactivity? Does it inhibit T-cell proliferation? Is it not recognized by NK cells? Is there experimental evidence at the bench that this cell line does not present antigens? Does the expression of PDL1 deliver an inhibiting signal in PD1+ T cells?

Author Response

The authors should analyze whether this cell line (compared with the others reported in the manuscript) is immunosuppressive. Does it not induce alloreactivity? Does it inhibit T-cell proliferation? Is it not recognized by NK cells? Is there experimental evidence at the bench that this cell line does not present antigens? Does the expression of PDL1 deliver an inhibiting signal in PD1+ T cells?

 We thank the reviewer for the comment and the supporting questions. We included an additional section with new Figure 7 from line 581 and included the findings in the text and the discussion section. Therefore, we included also a new experiment in the manuscript. We improved the title from “immunosuppressive” to “evasive”. We hope to give a clear picture on the immune evasive  nature of the LC4 liver cancer cells:

LC4 cells present a lower susceptibility towards active allogenic T-cells and cytolytic NK cells in vitro

To better understand and confirm the findings acquired by next generation sequencing followed by in-deep pathway analysis, we performed a proof-of-concept by treating the LC 4 cells on the one hand with activated T cells from an allogenic donor with HLA-A02:01 and on the other hand with immortalized activated NK92 cells. For a direct comparison and as positive control, we employed HepG2.1.3 cells which display the same haplotype as LC4 cells for the T cell transfer and Hep3B cells as MICA/B expressing cell line for the NK92 cell transfer. Both experiments were performed using the same settings, cell numbers, activation times and treatment time points, as displayed in Figure 7A – B. In Figure 7C and E the transfer of active allogenic T cells was observed over a period of 4 days. The highly significant reduction of the cell index was only measurable when transferring T cells to the susceptible control cell line HepG2.1.3. The cytolytic effect was even stronger after additional application of IL-2 after 24h, as shown in Figure 7C. For LC4 cells, we observed a much lower cytolytic effect, even when application of exogenous IL-2 was performed, as shown in Figure 7E. A other immune cell transfer experiment using LC4 cells was displayed in Figure 7B, D, F. NK92 cells were used in different E:T Ratios for the transfer to the susceptible HCC cell line Hep3B and to LC4 cells. The cytolytic effect was highly induced when applying 1:1 E:T NK92 cells on Hep3B cells, while this was not the case, when treating the LC4 cells. Here NK92 cells exhibit a strong cytolytic effect in higher E:T Ratios, like 1:10 and 1:20 E:T Ratios but not in 1:1 E:T Ratio, as shown in Figure 7F.

Round 2

Reviewer 2 Report

Comments and Suggestions for Authors

The authors addressed reviwers' comments

Author Response

We thank the reviewers for the previous comments and the opportunity to improve our work. 

Reviewer 3 Report

Comments and Suggestions for Authors

The authors showed that the LC4 cell line is growing less in the presence of allogeneic PBMC or the NK92 cell line. This effect is less evident than that observed with another HCC cell line, HepG2.1.3. I understand the effort of the authors in trying to further reinforce the idea that LC4 is somehow less sensitive to immune recognition. 

However, LC4 is proliferating more than other HCC cell lines( although the authors did not show this directly), and the less anti-proliferating effect shown is somewhat expected. The right experiments could be to see the perforin release and killing in short and long time (4h and 24.48h) exerted by immune cells instead of evaluating the cellular impedance with the xcelligence instrument. if this reviewer is right, the instrument evaluates the growth as a function of increased resistance due to cell attachment to the peculiar culture plates used. 

In addition, it is unclear whether the authors used adherent cells or spheroids. As spheroids are more similar (as stating by the authors) to original tumor it couldbe of great interst to understand the behavior of spheroids versus conventional cultures.

Anyway, the authors do a good job regarding the rest of the manuscript. I would suggest further down modulating the assertion that the LC4 "makes them an essential tool," line 765-766. I understand that the generation of different cell lines can help very well to understand the biology of HCC, but in this manuscript, the authors do not show direct experimental evidence of their pros versus the other cell lines of HCC.

Likewise, I suggest showing images of 2D and 3D cultures of LC4 to demonstrate the features of the two types of cultures. Indeed, they found differences in the two ways of culturing cells. This reinforces the good work of the authors in trying to characterize the present described cell line.

Finally, some of the suggested experiments could be inserted as limitations or better as potential experiments to clarify the immune behavior of the LC4 cell line.

Author Response

The authors showed that the LC4 cell line is growing less in the presence of allogeneic PBMC or the NK92 cell line. This effect is less evident than that observed with another HCC cell line, HepG2.1.3.

We thank the reviewer for the comment. The LC4 cells indeed reach higher cell index numbers than Hep3B and HepG2.1.3 cells when comparing Figures 7D and F or C and E in the non-treated proportion. We chose this graphical analysis instead of ratio calculation to give a transparent picture. For the interpretation, the untreated controls were plotted in graphs. However, we showed a clear event of cell lysis when comparing the NK cell-treated Hep3B cells on day 3 with their untreated counterparts.

On the other hand, the comparison between LC4 cells on day 3 treated with NK cells and their untreated controls doesn’t show the same lysis events. Even a high amount of NK cells (1:20 T:E) does not lead to a cytolytic event in the LC4 treatment. Only the proliferation is inhibited using NK cells, whereas in Figure 7D, we show an apparent reduction in the cell index resulting from NK cell killing in this setting. Additionally, we observed an evident ratio-related effect in both cell lines for the NK cell treatment, indicating that the recognition of the target cells is functioning; however, NK cell-mediated killing was suppressed in the LC4 cells.

For the T cells, we demonstrated that the same T cells from the same donor could significantly inhibit the proliferation of HCC cells. Furthermore, the application of IL-2 resulted in the lysis of HepG2.1.3 cells, as shown in Figure 7C. On the other hand, there was almost no evidence for the inhibition or lysis when applying T cells +/- IL2 to LC4 cells for 4 days, as shown in Figure 7E.

I understand the effort of the authors in trying to reinforce further the idea that LC4 is somehow less sensitive to immune recognition. However, LC4 is proliferating more than other HCC cell lines (although the authors did not show this directly), and the less anti-proliferating effect shown is somewhat expected. The right experiments could be to see the perforin release and killing in short and long time (4h and 24.48h) exerted by immune cells instead of evaluating the cellular impedance with the Xcelligence instrument. if this reviewer is right, the instrument evaluates the growth as a function of increased resistance due to cell attachment to the peculiar culture plates used. 

The Xcelligence measurement is accepted as a method to acquire cell lysis. In our setting, we wanted to show the co-culture-related killing of HCC cells. We used a gold standard cell line from NK cells, the Hep3B cells, as they express MICA/B, and the HepG2.1.3 cell for T cell transfer, since it presents the same HLA-A type as the LC4 cells.

In addition, it is unclear whether the authors used adherent cells or spheroids. As spheroids are more similar (as stated by the authors) to the original tumor, it could be of great interest to understand the behavior of spheroids versus conventional cultures.

We thank for the comment and clarified this point in the methods: “LC4 cells, Hep3B or HepG2.1.3 cells were seeded for 2-dimensional culture into Xcelligence (Agilent) 16-well plates (10.000/ well) including a negative medium control and were allowed to growth for 24h.(line126) and the Figure Legend: “ HCC cells and LC4 cells were seeded in 2D culture.” (line 603).  We also include: “Statistical outputs are indicated in the figure legends; p-values were plotted in the graph ****p ≤ 0.0001.”

Anyway, the authors do a good job regarding the rest of the manuscript. I would suggest further down modulating the assertion that the LC4 "makes them an essential tool," line 765-766. I understand that the generation of different cell lines can help very well to understand the biology of HCC, but in this manuscript, the authors do not show direct experimental evidence of their pros versus the other cell lines of HCC.

We thank again for making the point here. We want to reiterate that our LC4 cells were not immortalized. They provide a clinical picture within the context of CYP regulation, and they demonstrate a high preservation of parental behavior in terms of both toxic and canonical pathways. We still believe that immortalized cell lines are outdated, especially in the very individual therapy strategies we will focus on in the future. Therefore, cell lines like the LC4 are essential for the research community and the CRO work to approve new medications. We replaced “essential” tovaluable pre-clinical tool” (line 766-767) to clarify.

Likewise, I suggest showing images of 2D and 3D cultures of LC4 to demonstrate the features of the two types of cultures. Indeed, they found differences in the two ways of culturing cells. This reinforces the good work of the authors in trying to characterize the present described cell line.

We do show the pictures of 3D culture in Supplementary Figure 1 and published the pictures of the LC4 cells in the “Staffeldt, L., et al., Generating Patient-Derived HCC Cell Lines Suitable for Predictive In Vitro and In Vivo Drug Screening by Orthotopic Transplantation. Cells, 2023. 13(1).”

Finally, some of the suggested experiments could be inserted as limitations or better as potential experiments to clarify the immune behavior of the LC4 cell line.

We thank the reviewer again for the suggestion of the perforin acquisition after the transfer of immune cells. We performed a GzmB ELISA on the supernatant on day 4 and included the results of the measurements here. We observed constitutive secretion of GzmB in both co-culture experiments, as illustrated below.

Moreover, the level of GzmB secretion was related, as expected, to the T:E ratios. Interestingly, the level of GzmB was higher in the LC4 co-culture than in the Hep3B co-culture, showing less lysis of the target cells. Therefore, we obtain a clearer picture of the ability of LC4 cells to evade immune cell attraction, as seen in the secretion of GzmB. We did not include these findings in the manuscripts but wanted to provide them to Reviewer 3.

For clarification: The supernatant of each treatment group (n = 3) was pooled, and n = 2 technical replicates were measured; the experiment was performed twice. Each bar has n = 4 samples.
